# Epstein-Barr virus-driven B cell lymphoma mediated by a direct LMP1-TRAF6 complex

Fabian Giehler[1,2,3], Michael S. Ostertag[4], Thomas Sommermann[5], Daniel Weidl[6], Kai R. Sterz [2], Helmut Kutz[2], Andreas Moosmann[2,3,7], Stephan M. Feller[8], Arie Geerlof[4], Brigitte Biesinger[6], Grzegorz M. Popowicz[4], Johannes Kirchmair [9,10] & Arnd Kieser [1,2,3] ✉

Epstein-Barr virus (EBV) latent membrane protein 1 (LMP1) drives viral B cell transformation and oncogenesis. LMP1's transforming activity depends on its C-terminal activation region 2 (CTAR2), which induces NF-κB and JNK by engaging TNF receptor-associated factor 6 (TRAF6). The mechanism of TRAF6 recruitment to LMP1 and its role in LMP1 signalling remains elusive. Here we demonstrate that TRAF6 interacts directly with a viral TRAF6 binding motif within CTAR2. Functional and NMR studies supported by molecular modeling provide insight into the architecture of the LMP1-TRAF6 complex, which differs from that of CD40-TRAF6. The direct recruitment of TRAF6 to LMP1 is essential for NF-κB activation by CTAR2 and the survival of LMP1-driven lymphoma. Disruption of the LMP1-TRAF6 complex by inhibitory peptides interferes with the survival of EBV-transformed B cells. In this work, we identify LMP1-TRAF6 as a critical virus-host interface and validate this interaction as a potential therapeutic target in EBV-associated cancer.

The human gammaherpes virus EBV infects and transforms human B cells[1,2]. A global burden of approximately 164,000 deaths per year is associated with cancers driven by latent EBV infection, including cases of Burkitt's lymphoma (BL), Hodgkin's lymphoma (HL), EBV-positive diffuse large B cell lymphoma (DL-BCL), posttransplant lymphoproliferative disease (PTLD), as well as gastric and nasopharyngeal carcinoma (NPC)[3,4]. Of the viral proteins expressed during latency, only the oncoprotein LMP1 has the potential to transform rodent fibroblasts, and it is essential for viral B cell transformation into lymphoblastoid cell lines (LCLs)[5–9]. Expressed in the B cell compartment of mice, LMP1 causes fatal lymphoma if T cell-mediated immune surveillance is suppressed at the same time[10,11]. LMP1 is expressed in HL, DL-BCL, PTLD and NPC, where it critically contributes to pathogenesis[1,12].

LMP1 is a transmembrane protein of 386 amino acids that consist of a short N-terminus, six transmembrane domains and a C-terminal cytoplasmic signalling domain[13]. By spontaneous clustering in the membrane, LMP1 mimics a constitutively active receptor of the tumor necrosis factor (TNF) receptor family[8,14]. Essential for cell transformation by LMP1 are the C-terminal activation regions (CTARs) 1 and 2, which reside within the signalling domain[13]. Individual inactivation of either region greatly diminishes the transforming potential of EBV in primary B cells[9].

[1]Research Unit Signaling and Translation, Helmholtz Center Munich - German Research Center for Environmental Health, 85764 Neuherberg, Germany. [2]Research Unit Gene Vectors, Helmholtz Center Munich - German Research Center for Environmental Health, 81377 Munich, Germany. [3]German Center for Infection Research (DZIF), Partner Site Munich, Munich, Germany. [4]Institute of Structural Biology, Helmholtz Center Munich - German Research Center for Environmental Health, 85764 Neuherberg, Germany. [5]Immune Regulation and Cancer, Max Delbrück Center for Molecular Medicine, 13125 Berlin, Germany. [6]Institute of Clinical and Molecular Virology, University Hospital Erlangen, Friedrich-Alexander-University Erlangen-Nuremberg, 91054 Erlangen, Germany. [7]Department of Medicine III, University Hospital, Ludwig-Maximilians-University Munich, 81377 Munich, Germany. [8]Institute of Molecular Medicine, Martin-Luther-University Halle-Wittenberg, 06120 Halle, Germany. [9]Universität Hamburg, Department of Informatics, Center for Bioinformatics (ZBH), 20146 Hamburg, Germany. [10]Department of Pharmaceutical Sciences, Division of Pharmaceutical Chemistry, University of Vienna, 1090 Vienna, Austria. ✉e-mail: arnd.kieser@helmholtz-munich.de

CTAR1 directly recruits the TNF receptor-associated factor (TRAF) family members TRAF1, 2, and 3 via its TRAF-binding consensus motif $P_{204}xQxT$ to induce noncanonical and atypical NF-κB signalling, phosphoinositide 3-kinase (PI3K), and the mitogen-activated protein kinase (MAPK) pathways ERK and p38 (Fig. 1a)[15–21]. Furthermore, presumably indirect recruitment of TRAF5 and TRAF6 to CTAR1 has been reported[18,22]. In the presence of high levels of TRAF1, CTAR1 also contributes to canonical NF-κB and c-Jun N-terminal kinase (JNK) activation[23–27].

CTAR2, the major activation site of canonical NF-κB, JNK, p38, and interferon regulatory factor 7 (IRF7), recruits TRAF6 as the critical signalling mediator for all known CTAR2-induced pathways[13,20,22,28–31]. TRAF6 bridges CTAR2 with the downstream mediators TNIK (TRAF2- and NCK-interacting protein kinase), TAK1 (transforming growth

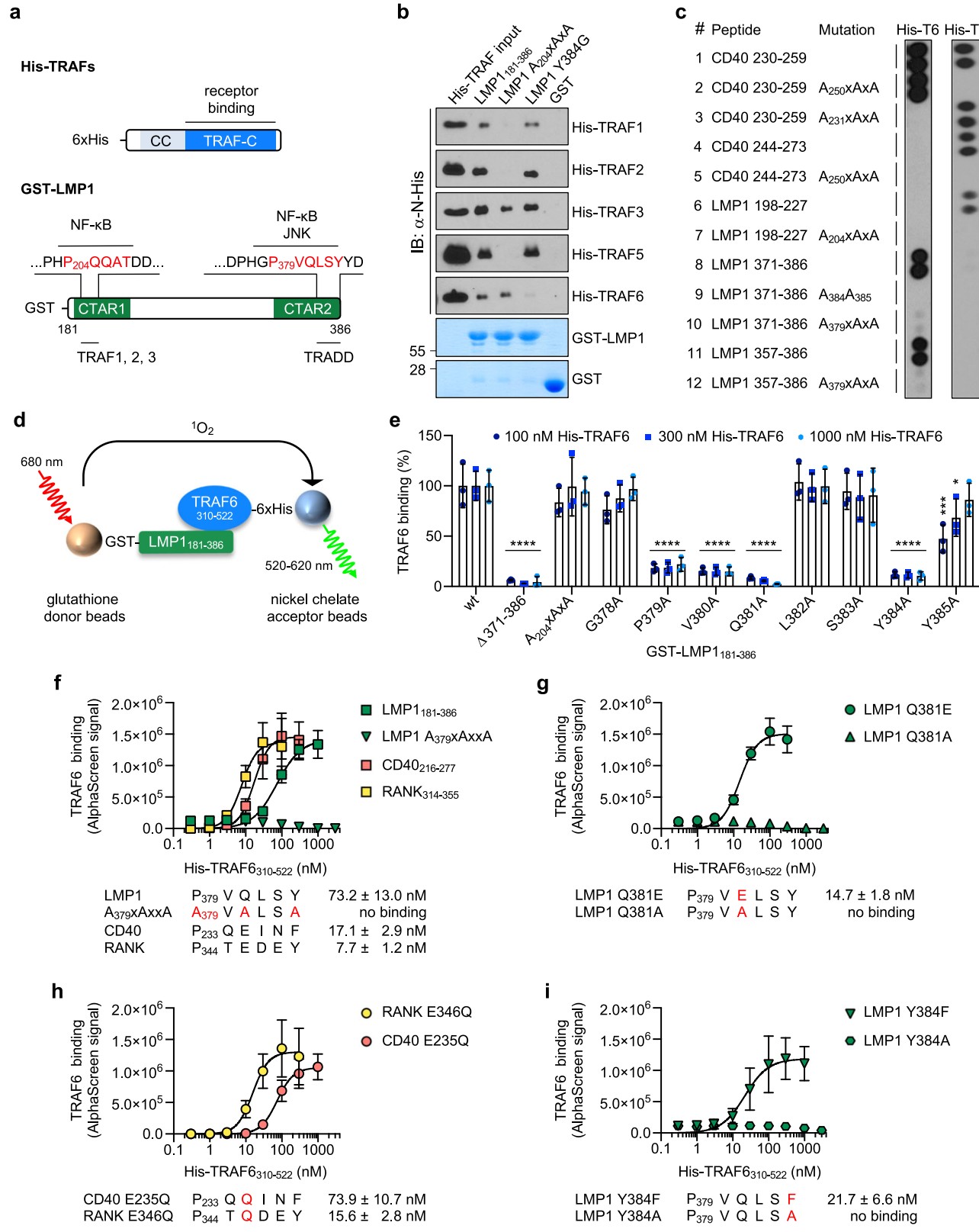

**Fig. 1 | Direct interaction of TRAF6 with the CTAR2 of LMP1. a** Schematic depiction of recombinant His-tagged TRAF and GST-LMP1 proteins used in this study. The CTAR1 and CTAR2 core sequences are highlighted in red. Relevant previously described direct interaction partners are shown. CC, coiled coil. **b** TRAF1, 2, 3, and 5 interact with the PxQxT motif of CTAR1, whereas TRAF6 directly binds to the CTAR2 domain. Recombinant His-TRAF proteins were detected via their His-tags on immunoblots (IB) of pulldowns with the indicated GST-LMP1 proteins. Uncropped blots with molecular weights in Supplementary Fig. 6a. Representative results are shown. Number of independent experiments: TRAF1, $n = 2$; TRAF2, $n = 3$; TRAF3, $n = 3$; TRAF5, $n = 3$; TRAF6, $n = 3$. **c** Mutation of the CTAR2 sequence $P_{379}$VQLSY abolishes the TRAF6 interaction with LMP1. LMP1-derived peptides and CD40-derived controls were immobilized on membranes in duplicate and incubated with recombinant His-TRAF6$_{310-522}$ (His-T6) or His-TRAF2$_{311-501}$ (His-T2). Peptide binding of TRAF6 or TRAF2 proteins was detected by

TRAF-specific antibodies. The data are representative of two independent experiments. **d** Design of the AlphaScreen PPI assay for the detection and quantification of direct TRAF6 binding to LMP1. **e** LMP1 residues $P_{379}$, $V_{380}$, $Q_{381}$ and $Y_{384}$ are essential for TRAF6 binding to LMP1 in AlphaScreen PPI assays. Data are mean values ± standard deviation (SD) of three independent experiments. Statistics: two-way ANOVA. $p$-values: *$p \leq 0.05$, ***$p \leq 0.001$, ****$p \leq 0.0001$. Source data and exact $p$-values in the Source Data file. **f–i** Quantitative analysis of TRAF6 interaction with LMP1, CD40, and RANK. His-TRAF6$_{310-522}$ was tested at different concentrations in AlphaScreen PPI assays with GST-LMP1$_{181-386}$, GST-CD40$_{216-277}$, GST-RANK$_{314-355}$, and the indicated mutants. $K_D$ values are given for measurable interactions. Data are mean values ± SD of three (CD40), four (LMP1), or six (RANK) independent experiments. Curve fitting: Prism, one site-specific binding with hill slope. Source data in the Source Data file.

factor β-activated kinase 1), TAB1/2 (TGFβ-activated kinase 1 binding protein 1/2), the IKK (IκB kinase) complex, and LIMD1 (LIM domain-containing protein 1)[32–35]. In contrast to the TNF-receptor-associated death domain protein (TRADD), which binds to the sixteen C-terminal amino acids of CTAR2 and is involved in NF-κB signalling by LMP1, the mechanism of TRAF6 recruitment and, thus, the molecular basis for its essential role in LMP1 signalling remains unknown[31,36–39].

The sequence $P_{379}$VQLSY$_{384}$, located at the C-terminus of CTAR2, is responsible for NF-κB and JNK activation by CTAR2 (Fig. 1a)[38,40]. Although this sequence includes the putative TRAF1/2/3 binding motif PxQxS[41], direct physical interaction of CTAR2 with TRAF molecules has never been demonstrated. One study suggested that TRAF6 recruitment to CTAR2 might be indirect, possibly mediated by the transcription factor BS69[42].

The TRAF protein family consists of seven members, TRAF1 to 7. Of these seven isoforms, TRAF1 to 6 share the so-called TRAF (or MATH) domain, which is located at the C-terminus of the TRAF molecule and is composed of a TRAF-N (or coiled-coil) domain and a TRAF-C domain, the latter built of seven to eight anti-parallel β-strand folds[43,44]. Mediated by their TRAF domains, TRAF proteins form mushroom-like trimers, which can interact with receptors through TRAF-C[44–46]. The TRAF domain of TRAF6 is sufficient to mediate TRAF6 interaction with the LMP1 complex[20]. TRAF6 lacking its TRAF domain is unable to rescue TRAF6 deficiency in LMP1 signalling, further supporting a role of this domain for interaction with LMP1[22]. The N-terminal RING finger of TRAF6 possesses E3 lysine 63 (K63)-linked ubiquitin ligase activity, which plays an important role in the activation of LMP1 downstream signalling including TAK1 and IKKβ activation[13,47].

LMP1 mimics signals of the costimulatory receptor CD40, a member of the TNF receptor family, during B cell proliferation and can largely replace CD40 functions in vivo[8,48–51]. TRAF6 deficiency affects B cell numbers driven by a conditional CD40-LMP1 fusion protein in the lymph nodes of mice[52]. However, a potential role of TRAF6 in LMP1-dependent lymphoma has not been demonstrated. Although both LMP1 and CD40 engage TRAF6 in signalling, the underlying molecular mechanisms seem to differ. CD40 carries two major TRAF binding sites, a TRAF1/2/3 binding site with the sequence $P_{250}$VQET and the TRAF6 interaction site $Q_{231}$EPQ$_{235}$EINF[53,54]. The TRAF binding sites of CD40 are largely redundant with respect to their functions in NF-κB and JNK activation in B cells[55]. At the molecular level, JNK signalling induced by LMP1 differs from that induced by CD40 regarding the functions of IKKβ and TPL2[35,56].

In this work, we characterize the interaction between LMP1 and TRAF6 as a virus-host interface, which is based on direct protein-protein interaction. We provide structural insights into the molecular architecture of the LMP1-TRAF6 complex and demonstrate that the direct interaction of LMP1 and TRAF6 is critical for LMP1 function and the survival of EBV-transformed B cells. In summary, we reveal the molecular mechanism of TRAF6 engagement by LMP1 for signalling and lymphoma development.

## Results

### TRAF6 interacts directly with $P_{379}$VQLSY$_{384}$ of LMP1

We examined all TRAF proteins involved in LMP1 signalling regarding their potential to directly bind to the LMP1 signalling domain (Fig. 1a and b, Supplementary Fig. 1a). The purified recombinant TRAF domains of TRAF1, 2, 3, and 5 interacted with $P_{204}$QQAT of CTAR1 in pull-down assays with glutathione S-transferase (GST)-coupled LMP1$_{181-386}$ (Fig. 1b). Mutation of $P_{204}$xQxT into $A_{204}$xAxA abolished LMP1 binding to TRAF1, 2, and 5. Residual amounts of TRAF3 were recruited by the $A_{204}$xAxA mutant, which can be explained by the contacts of TRAF3 with LMP1 residues adjacent to the $P_{204}$xQxT core motif[57].

Investigating the interaction between LMP1 and TRAF6, we made the surprising observation that recombinant His-TRAF6$_{310-522}$, which includes the TRAF domain of TRAF6, was efficiently recruited by GST-LMP1 in this two-component system (Fig. 1b). In contrast to all other TRAF proteins tested, TRAF6 recruitment to LMP1 was not affected by mutation of CTAR1 but was eliminated by the exchange of $Y_{384}$ for glycine. In accordance with this finding, wild-type Flag-TRAF6 only coimmunoprecipitated with HA-LMP1 from HEK293 cells if CTAR2 was intact (Supplementary Fig. 1b). These experiments provided evidence for a direct protein-protein interaction (PPI) as the molecular basis of TRAF6 recruitment to LMP1.

To further substantiate this observation and to narrow down the LMP1 sequences that are involved in the direct LMP1-TRAF6 interaction, we tested the ability of purified His-TRAF6$_{310-522}$ to interact with immobilized LMP1-derived peptides, which incorporate CTAR1 or CTAR2 sequences (Fig. 1c and Supplementary Fig. 1c). Recombinant His-TRAF2$_{311-501}$ was used as a control. The specificity of the TRAF interaction was confirmed by including peptides that harbored alanine exchanges within the TRAF2-binding motifs of CD40 ($P_{250}$VQET to $A_{250}$VAEA, peptides 1 and 2, respectively), LMP1 ($P_{204}$QQAT to $A_{204}$QAAA, peptides 6 and 7), and the TRAF6 binding motif of CD40 ($Q_{231}$EPQEINF to $A_{231}$EAQAINF, peptides 1 and 3). CD40-derived amino acids 244-273 lack the TRAF6 binding site (peptide 4). Additional mutation of the TRAF2-binding motif within peptide 4 resulted in peptide 5. Previously, we showed that amino acids 371-386 of LMP1 are sufficient to induce TRAF6-dependent CTAR2 signalling[31]. To determine whether these sixteen amino acids contain the complete TRAF6 binding site of LMP1, we included them as peptide 8. Within peptide 8, $Y_{384}$ and $Y_{385}$ or the cryptic TRAF interaction motif $P_{379}$xQxS were mutated (peptides 9 and 10, respectively). CTAR2 amino acids 357-386 were spotted (peptide 11), in which $P_{379}$xQxS was mutated (peptide 12).

Both TRAF2 and TRAF6 specifically interacted with their designated binding sites within CD40, confirming the validity of the peptide array (Fig. 1c). Furthermore, TRAF2 bound to $P_{204}$QQAT of CTAR1 (peptides 6 and 7) but not to CTAR2 (peptides 8 to 12), which excludes the possibility of a direct TRAF2 interaction with the cryptic TRAF interaction motif of CTAR2. TRAF6, however, was efficiently captured by CTAR2 peptides 8 (16mer) and 11 (30mer), whereas it did not bind to

CTAR1 (peptide 6). Mutation of $Y_{384}$ and $Y_{385}$ to AA (peptide 9) and $P_{379}xQxS$ to AxAxA (peptides 10 and 12) abolished the direct TRAF6 binding to CTAR2 (Fig. 1c).

We performed an alanine exchange mutagenesis scan from $G_{378}$ to $Y_{385}$ of LMP1 to precisely map the residues involved in TRAF6 binding. We developed a highly reliable mix-and-measure screening assay for the LMP1-TRAF6 interaction based on the AlphaScreen technology (Perkin Elmer), by which the effects of mutations on this protein-protein interaction can be detected and quantified directly (Fig. 1d). In this assay, the light emission at 520-620 nm is directly proportional to the affinity of the two protein components of the assay. Each of the LMP1 amino acids $P_{379}$, $V_{380}$, $Q_{381}$ and $Y_{384}$ was essential for the direct TRAF6 recruitment to the LMP1 signalling domain (Fig. 1e). Mutation of $Y_{385}$ had only a minor impact on TRAF6 binding at the lower TRAF6 concentrations tested in the assay, whereas the side chains of $G_{378}$, $L_{382}$ and $S_{383}$ were irrelevant for interaction. Of note, the resulting TRAF6 binding sequence $P_{379}VQxxY$ exactly matches the NF-κB- and JNK-inducing region of CTAR2[38,40]. This finding strongly suggested that the direct binding of TRAF6 to this sequence is, in fact, the molecular basis for CTAR2 signalling.

LMP1 showed a weaker affinity for TRAF6 than the cellular TRAF6-interacting receptors CD40 and receptor activator of NF-κB (RANK, also known as TRANCE receptor) (Fig. 1f). The $K_D$ values of the His-$TRAF6_{310-522}$ interaction with GST-CD40 and GST-RANK were $17.1 \pm 2.9$ nM and $7.7$ nM $\pm 1.2$ nM, respectively, in contrast to $73.2 \pm 13.0$ nM with GST-LMP1, as determined by the AlphaScreen PPI assay. Confirming our previous data, mutation of LMP1 $P_{379}xQxxY$ into $A_{379}xAxxA$ abolished TRAF6 binding. The analogous mutation of the TRAF6 binding motif within CD40, which was included as a control, also resulted in a loss of TRAF6 interaction (Supplementary Fig. 1d).

Alignment of the consensus TRAF6 interaction motif PxExxF/Y/D/E of cellular receptors[46,53] with the newly identified viral TRAF6 binding sequence of LMP1 revealed a high degree of similarity, with the exception of one striking difference at the central position $P_0$ (Supplementary Fig. 1e). All known cellular TRAF6-recruiting sequences carry a glutamic acid at $P_0$[46,58–60], whereas in LMP1 this position is occupied by glutamine. We tested the effect of converting the TRAF6 binding site of LMP1 into the cellular consensus motif by exchanging $Q_{381}$ at $P_0$ for glutamic acid (Fig. 1g). TRAF6 was capable of binding the resulting LMP1 $Q_{381}E$ mutant with 5.0-fold enhanced affinity ($14.7 \pm 1.8$ nM), which is in the range of its affinity to CD40 and RANK (Fig. 1g, compare to 1f). Previous work with peptide arrays suggested that glutamic acid at $P_0$ of the TRAF6 binding motif of CD40 cannot be replaced by any other amino acid, including glutamine, without the loss of affinity towards TRAF6[58]. We examined in our quantitative PPI assay whether the exchange of glutamic acid for glutamine at $P_0$ of the cellular receptors eliminates their affinity towards TRAF6. E to Q mutation at $P_0$ did not abolish the affinity of RANK to TRAF6 but reduced it by a factor of 2.0. Likewise, the E-Q exchange reduced the affinity of CD40 to TRAF6 by a factor of 4.3 (Fig. 1h, compare to 1f). Based on these observations we concluded that a Q-E exchange at $P_0$ has similar effects on all tested TRAF6 interaction motifs and that there is no fundamental structural difference between the viral and cellular motifs at $P_0$. Our results suggest PxQ/ExxF/Y/D/E as the new extended consensus TRAF6 interaction motif.

Position $P_3$ of LMP1's TRAF6-binding motif is filled by $Y_{384}$. This LMP1 residue has a critical role in LMP1 signalling and viral cell transformation[9,20,38,40]. In cellular TRAF6-interacting receptors this position can be occupied by an aromatic or acidic amino acid[46]. To test variability at $P_3$ of LMP1, we introduced a $Y_{384}F$ mutation resembling $P_3$ of the TRAF6 binding motif of CD40. This mutation improved the affinity of LMP1 for TRAF6 (Fig. 1i).

Although CTAR1 harbors the putative TRAF6 interaction motif $P_{204}xQxxD_{209}$, we observed no direct interaction of TRAF6 with CTAR1 (Fig. 1b, c, e, and f). Furthermore, an $A_{204}xAxA$ mutation, which would destroy this putative TRAF6 site, had no effect on overall TRAF6

affinity to LMP1 (Fig. 1e and Supplementary Fig. 1f). The question arose why does CTAR1 not bind TRAF6. One obvious difference between the CTAR1 and CTAR2 motifs is that $P_3$ of CTAR1 is occupied with $D_{209}$ instead of $Y_{384}$ within CTAR2. We converted the TRAF6 interaction motif of CTAR2 into a CTAR1-like motif by inserting a $Y_{384}D$ mutation. $Y_{384}D$ caused a 1.8-fold reduced affinity of LMP1 towards TRAF6 but TRAF6 binding was not abolished (Supplementary Fig. 1f, compare to Fig. 1f). Therefore, the presence of aspartate at $P_3$ of CTAR1 does not explain its missing affinity towards TRAF6. Additional factors adjacent to the putative TRAF6 core motif of CTAR1 might thus prevent TRAF6 interaction.

Taken together, TRAF6 is directly recruited by the JNK- and NF-κB-inducing sequence $P_{379}VQLSY$ within CTAR2 and is, thus, the first identified cellular factor whose binding site matches exactly the signalling-active site of CTAR2. In contrast to cellular receptors, this motif contains glutamine at the central $P_0$ position.

### Arginine 392 of TRAF6 discriminates between LMP1 and CD40

To examine whether LMP1 binds to the same region at the surface of TRAF6 as cellular receptors, we mutated the amino acids within TRAF6 that are involved in the interaction with $P_{-2}$, $P_0$, or $P_3$ of CD40 and RANK[46]. The capability of the TRAF6 mutants $R_{392}A$, $K_{469}A$, $F_{471}A$, or $Y_{473}A$ to bind to GST-LMP1 and GST-CD40 was analyzed in AlphaScreen PPI experiments (Fig. 2a and b, respectively). $F_{471}$ and $Y_{473}$ of TRAF6 build the binding pocket for amino acid $P_{-2}$ of cellular receptors[46,60]. The substitutions $F_{471}A$ and $Y_{473}A$ caused a complete loss of TRAF6 binding to both LMP1 and CD40. Hence, this pocket forms an essential interaction with LMP1, most probably with LMP1 residue $P_{379}$, which occupies $P_{-2}$ of the $P_{379}VQxxY$ motif. The $Y_{471}A$ mutation also abolished the interaction of TRAF6 with RANK confirming a similar TRAF6 binding mode at $P_{-2}$ of RANK (Fig. 2c). Mutation of $K_{469}$ to alanine did not affect the TRAF6 interaction with LMP1 or CD40. The side chain of $K_{469}$ likely forms nonessential charge-charge interactions with the main chain carboxylate of $P_0$ of CD40[46].

Notably, the TRAF6 mutant $R_{392}A$ was found to discriminate between LMP1 and RANK on one side and CD40 on the other side (Fig. 2a–c). $R_{392}A$ mutation prevented the interaction of TRAF6 with LMP1 and RANK. Even LMP1 $Q_{381}E$, which has a higher affinity towards TRAF6 than LMP1 wild-type (see Fig. 1i), was unable to override the negative effect of $R_{392}A$ mutation (and also of $F_{471}A$ mutation; Supplementary Fig. 2a). In contrast, $R_{392}A$ mutation had no impact on TRAF6 binding to CD40 (Fig. 2b), indicating that a potential amino-aromatic interaction of TRAF6 $R_{392}$ with $F_{238}$ at $P_3$ of CD40[46] is not essential. So far, our functional studies suggested that the molecular architecture of the LMP1-TRAF6 complex shows similarities to the RANK-TRAF6 complex, which both differ from the CD40-TRAF6 interaction, especially around $P_3$.

To verify the relevance of our findings for the LMP1-TRAF6 interaction in vivo, we expressed Flag-tagged TRAF6 wild-type or the mutants $R_{392}A$, $K_{469}A$, $F_{471}A$ and $Y_{473}A$ together with HA-tagged LMP1 in HEK293 cells and performed coimmunoprecipitations of both proteins (Fig. 2d). Confirming our previous results, each of the mutations $R_{392}A$, $F_{471}A$ or $Y_{473}A$, abolished the TRAF6 interaction with LMP1 in HEK293 cells, whereas the $K_{469}A$ mutation had no negative effect on the interaction between the two proteins. Confocal immuno-fluorescence studies in HeLa cells further verified these results (Fig. 2e). Flag-TRAF6 wild-type and the $K_{469}A$ mutant colocalized to a high extent with HA-LMP1 clusters, demonstrating their interaction with LMP1 in situ. In contrast, the TRAF6 mutants $R_{392}A$, $F_{471}A$ and $Y_{473}A$ showed a strongly decreased colocalization with HA-LMP1, which was comparable to the LMP1Δ371-386 mutant lacking the TRAF6 interaction site (Fig. 2e and Supplementary Fig. 2b). In the absence of LMP1, all TRAF6 mutants showed a similar cytoplasmic distribution as the wild-type (Supplementary Fig. 2c). In summary, these results demonstrated that binding of TRAF6 to LMP1 involves the same TRAF6

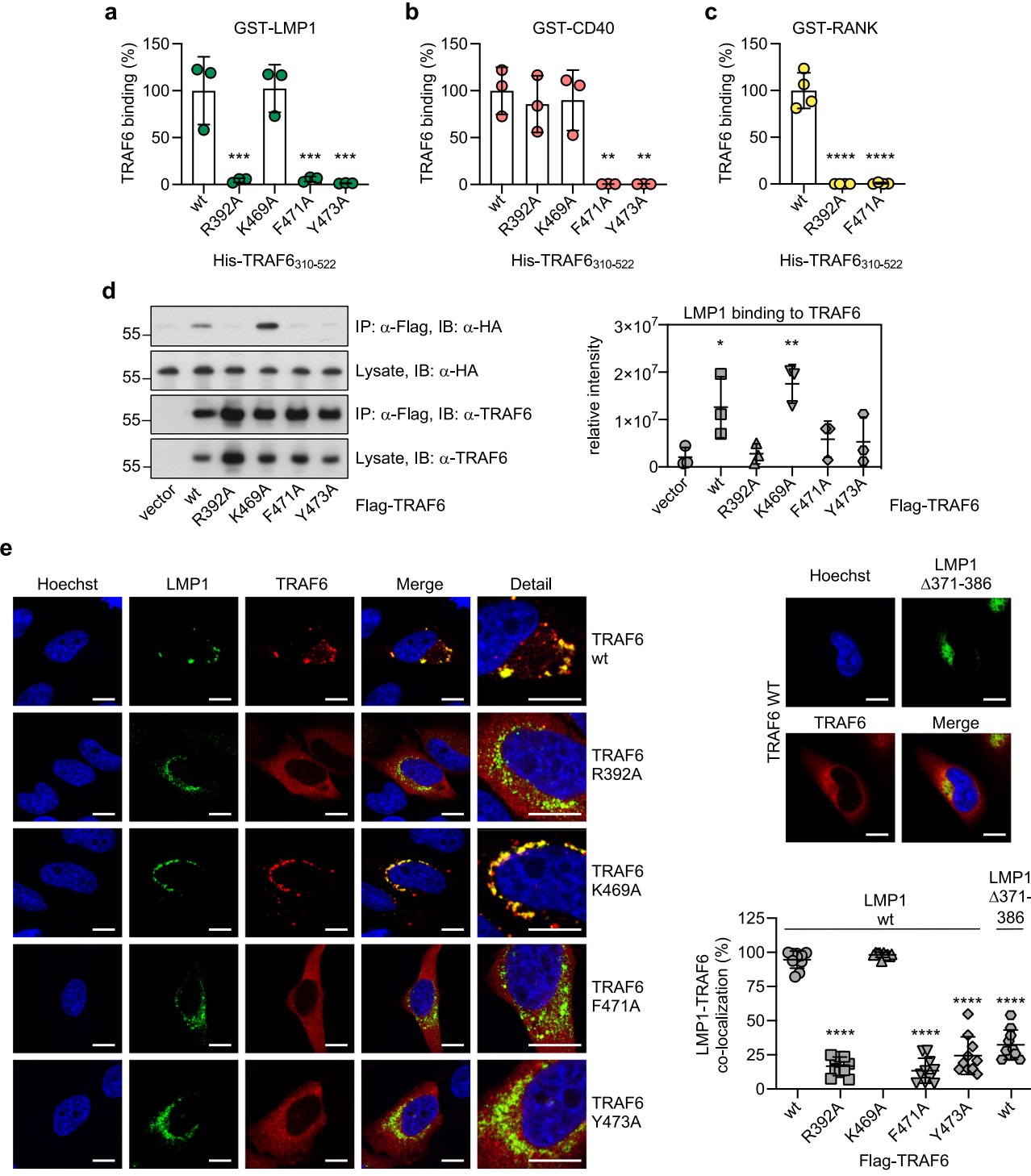

**Fig. 2 | LMP1 binds to the receptor-interacting surface of TRAF6. a** Mutations within the receptor-interacting surface of TRAF6 affect binding to LMP1. The TRAF6 mutants $R_{392}A$, $K_{469}A$, $F_{471}A$, and $Y_{473}A$ were tested LMP1 in AlphaScreen PPI experiments with GST-LMP1$_{181-386}$. Data are mean values ± SD of three independent experiments. Statistics: one-way ANOVA. **b** TRAF6 $R_{392}A$ differentiates between CD40 and LMP1/RANK. AlphaScreen PPI experiments with GST-CD40$_{216-277}$. Data are mean values ± SD of three independent experiments. Statistics: one-way ANOVA. **c** TRAF6 $R_{392}A$ and $Y_{471}A$ do not bind to RANK. AlphaScreen PPI experiments with GST-RANK$_{314-355}$. Data are mean values ± SD of four independent experiments. Statistics: one-way ANOVA. **d** TRAF6 mutants that fail to interact with LMP1 in PPI assays are also unable to bind cellular LMP1. HEK293 cells were co-transfected with HA-LMP1 and the indicated Flag-TRAF6 mutants. Flag-TRAF6 was immunoprecipitated (IP) via its Flag-tag and coprecipitated HA-LMP1 was detected on immunoblots by an α-HA antibody. Uncropped blots in Supplementary Fig. 6b. For statistical analysis LMP1 signals were digitalized and quantified (graph). Data are mean values ± SD of three independent experiments. Statistics: one-way ANOVA. **e** TRAF6 $R_{392}A$, $F_{471}A$, and $Y_{473}A$ fail to interact with LMP1 wild-type clusters in HeLa cells (large panel). TRAF6 recruitment is dependent on amino acids 371 – 386 of LMP1 (small panel). Confocal microscopy images show representative cells of three independent experiments. Quantitative data are mean values ± SD of ten randomly selected cells per transfection of one representative experiment (graph). Scale bars: 10 μm. Statistics: one-way ANOVA. *p*-values: *$p \leq 0.05$, **$p \leq 0.01$, ***$p \leq 0.001$, ****$p \leq 0.0001$. Source data and exact p-values in the Source Data file.

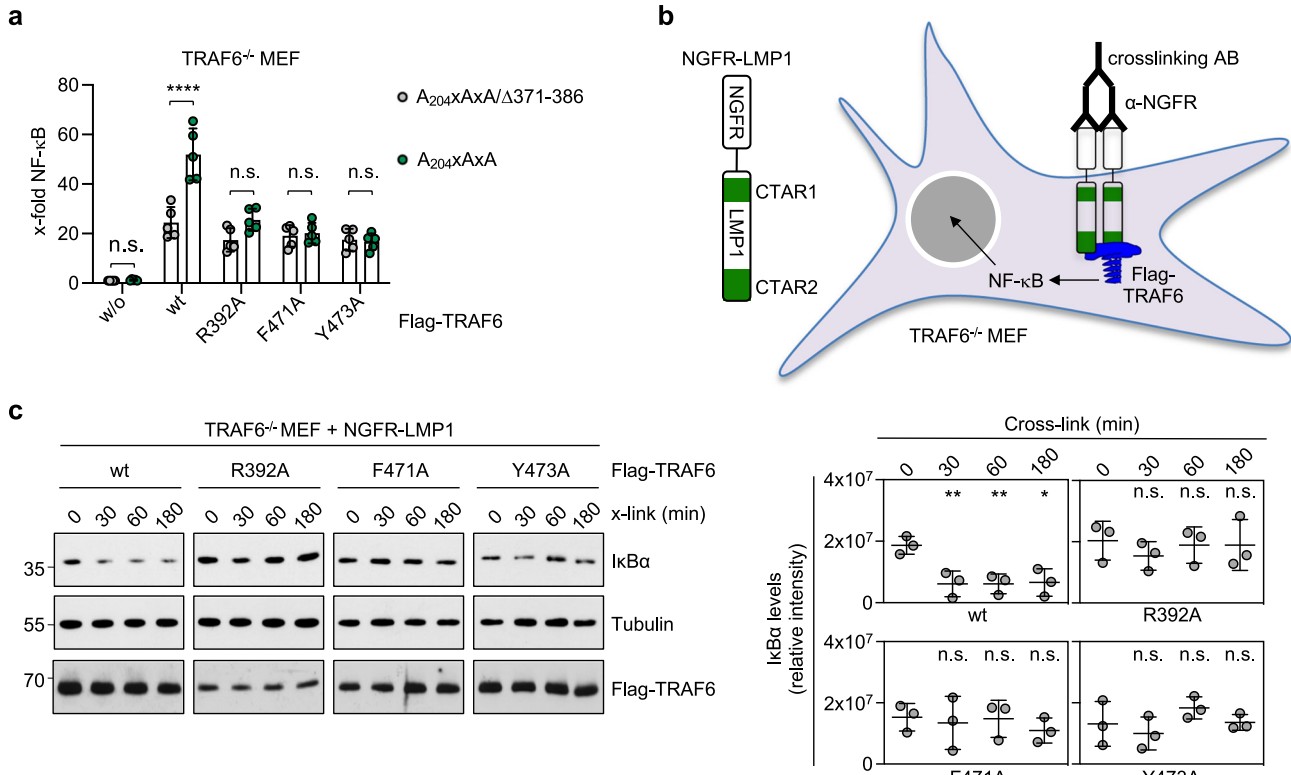

**Fig. 3 | Direct binding of TRAF6 to LMP1 is essential to activate signalling by CTAR2. a** TRAF6 mutants incapable of direct LMP1 binding fail to rescue CTAR2 signalling to NF-κB. Transient NF-κB reporter assays in TRAF6$^{-/-}$ MEFs. CTAR2-specific signalling was induced by the CTAR1 mutant A$_{204}$xAxA. The CTAR1/CTAR2 double mutant A$_{204}$xAxA/Δ371-386 served as inactive null control. TRAF6 wild-type or mutants were cotransfected as indicated. Data are mean values ± SD of five independent experiments. Statistics: two-way ANOVA. **b** NGFR-LMP1 consists of the LMP1 signalling domain and the transmembrane and extracellular domains of the p75 NGF receptor. NGFR-LMP1 activity is induced at the cell surface by an α-NGFR primary antibody and subsequent cross-linking with a secondary antibody,

allowing time-resolved analysis of LMP1 signalling. **c** TRAF6 mutants lacking direct LMP1 binding activity are unable to rescue canonical NF-κB activation by NGFR-LMP1 in TRAF6-deficient MEFs. TRAF6$^{-/-}$ MEFs stably expressing NGFR-LMP1 were transduced with Flag-TRAF6 wild-type or the indicated mutants, NGFR-LMP1 activity was induced by antibody crosslinking for the indicated times, and IκBα levels were analyzed by immunoblotting. Uncropped blots in Supplementary Fig. 6c. Representative blots are shown (left). Data are mean values ± SD of three independent experiments (right). Statistics: one-way ANOVA. p-values: *$p \leq 0.05$, **$p \leq 0.01$, ****$p \leq 0.0001$, n.s. (not significant). Source data and exact p-values in the Source Data file.

residues in the cellular context as in our interaction studies with recombinant protein, which strongly argues for the same and direct mechanism of LMP1-TRAF6 complex formation in vivo as in vitro.

## The direct binding of TRAF6 to LMP1 is required for CTAR2 signalling

CTAR2 signalling is defective in TRAF6-deficient mouse embryonic fibroblasts (MEFs) and can be rescued by exogenous TRAF6 expression[20,28,29,31]. To demonstrate that the direct interaction of LMP1 and TRAF6 is indeed the molecular basis for CTAR2 signalling, we tested the TRAF6 mutants that are defective in direct LMP1 binding for their potential to rescue CTAR2 signalling in NF-κB reporter assays in TRAF6$^{-/-}$ MEFs (Fig. 3a). TRAF6$^{-/-}$ cells were transfected with the CTAR1 mutant A$_{204}$xAxA, which signals towards NF-κB only through CTAR2, or the inactive double mutant A$_{204}$xAxA/Δ371-386, together with wild-type TRAF6 or the TRAF6 mutants R$_{392}$A, F$_{471}$A, or Y$_{473}$A. Comparable protein expression levels were confirmed by immunoblot analysis (Supplementary Fig. 3a). In the absence of TRAF6, CTAR2 was unable to induce NF-κB reporter activity (Fig. 3a, see w/o). As expected, expression of wild-type TRAF6 or the TRAF6 mutants alone (cotransfection with inactive A$_{204}$xAxA/Δ371-386) induced NF-κB to similar levels (grey values), demonstrating that all mutants fully retained their downstream signalling capacity. However, only TRAF6 wild-type, but none of the binding-defective mutants, was able to rescue CTAR2 signalling to NF-κB (green values). This result shows that the

direct interaction of TRAF6 with LMP1 is critical for the activation of CTAR2-mediated NF-κB signalling.

Next, we retrovirally transduced TRAF6$^{-/-}$ MEFs, which stably express NGFR-LMP1, with TRAF6 wild-type or the mutants R$_{392}$A, F$_{471}$A and Y$_{473}$A. NGFR-LMP1 is a conditional fusion construct of the extracellular and transmembrane domains of the p75 nerve growth factor (NGF) receptor (NGFR) with the intracellular signalling domain of LMP1[14,35]. Instant NGFR-LMP1 activity can be triggered at the cell surface by stimulation of the cells with an α-NGFR primary antibody and a crosslinking secondary antibody (Fig. 3b). NGFR-LMP1 activation caused a rapid degradation of IκBα, indicating the induction of the canonical NF-κB pathway in wild-type MEFs (Supplementary Fig. 3b)[35]. In TRAF6$^{-/-}$ cells, this pathway was defective (Supplementary Fig. 3b). Exogenous expression of TRAF6 wild-type in TRAF6$^{-/-}$ cells restored activation of the canonical NF-κB pathway upon NGFR-LMP1 crosslinking (Fig. 3c). In contrast, the TRAF6 mutants R$_{392}$A, F$_{471}$A and Y$_{473}$A, which are unable to directly bind to LMP1, were also ineffective in rescuing canonical NF-κB activation by CTAR2 (Fig. 3c). Taken together, these data demonstrate that CTAR2 only induces NF-κB if TRAF6 is directly recruited.

## Molecular modelling of the LMP1-TRAF6 complex

Our experiments with recombinant TRAF6 proteins provide evidence that LMP1 binds to the same PPI interface of TRAF6 as the cellular receptors CD40 and RANK (see Fig. 2). To gain a closer insight into the

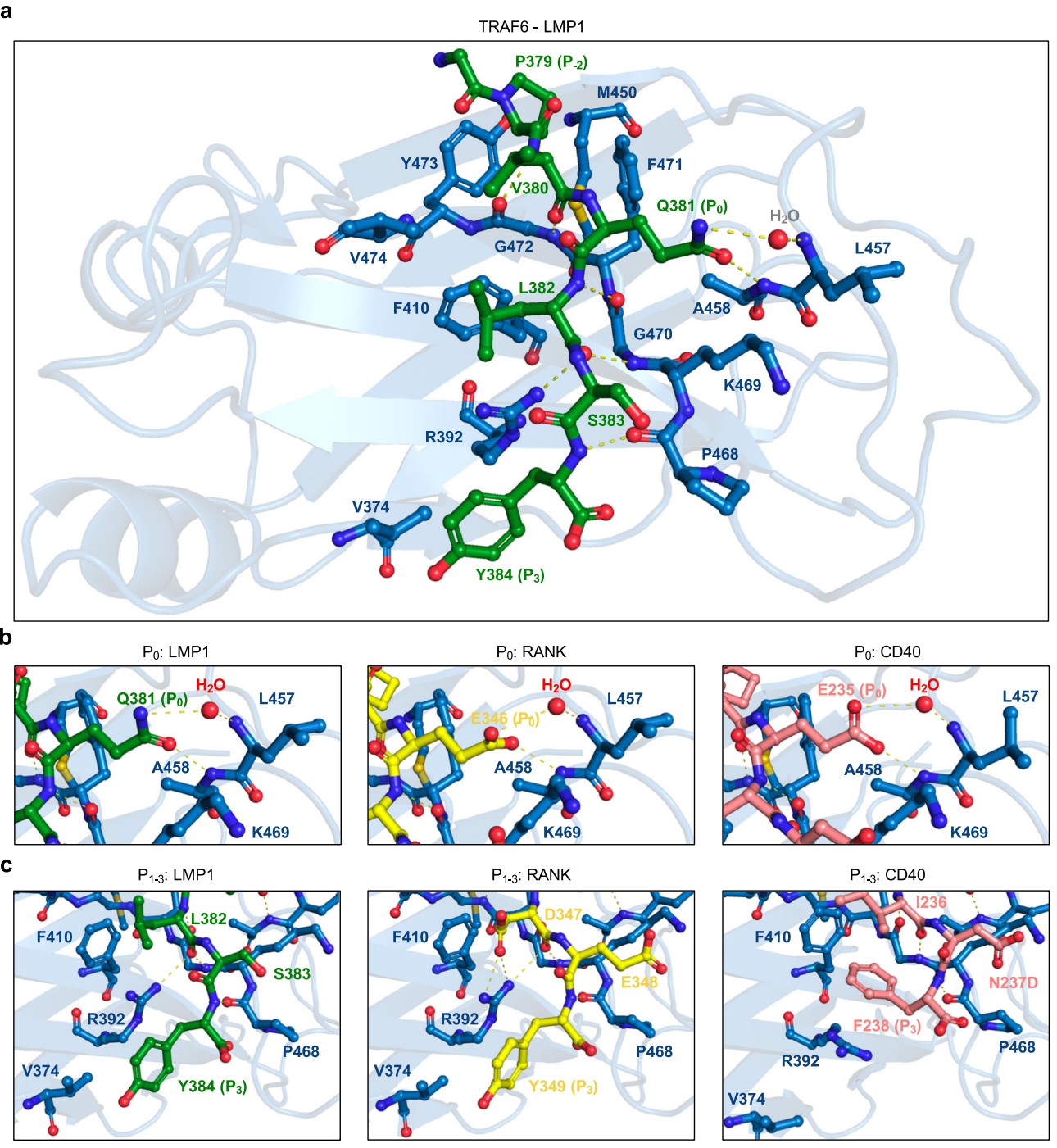

**Fig. 4 | Molecular model of LMP1 in complex with TRAF6. a** In silico structure of the LMP1 peptide $G_{378}PVQLSY_{384}$ (green) bound to the receptor-binding interface of the TRAF domain of TRAF6 (blue), as derived from a RANK-TRAF6 crystal structure (PDB 1LB5)[46]. Hydrogen bonds formed between LMP1 and TRAF6 are indicated by yellow, dashed lines. **b** TRAF6 environment around position $P_0$ of LMP1 (in silico model; green), compared to RANK (PDB 1LB5; yellow) and CD40 (PDB 1LB6; salmon). **c** TRAF6 environment around positions $P_1$-$P_3$ of LMP1 compared to RANK and CD40. The tyrosine at $P_3$ of RANK and LMP1 adopts a stretched conformation versus the kinked conformation of $F_{238}$ of CD40. Note that the mutation $N_{237}D$ was introduced into CD40 to enable crystallization[46].

structural properties of the LMP1-TRAF6 complex, we derived an in silico model of this interaction from a crystal structure of the RANK-TRAF6 complex (PDB 1LB5)[46] by mutating the relevant residues of RANK, with subsequent energy minimization (see Methods) (Fig. 4a). The RANK-TRAF6 structure was preferred over the CD40-TRAF6 structure PDB 1LB6[46] as the template for modelling, because our functional studies suggested that LMP1-TRAF6 is more closely related to RANK-TRAF6 than to CD40-TRAF6.

Position $P_{-2}$ of LMP1 and both cellular receptors is occupied by proline (see Supplementary Fig. 1e). According to the model, the proline of LMP1 at $P_{-2}$ ($P_{379}$) is located in the hydrophobic indentation formed primarily by TRAF6 residues $M_{450}$, $F_{471}$ and $Y_{473}$ (Fig. 4a). In line with this observation, mutation of the TRAF6 residues $F_{471}$ and $Y_{473}$ abolished LMP1 binding (see Fig. 2).

At $P_{-1}$ of LMP1, hydrogen bonds are formed between the main chain of $V_{380}$ and the main chain of TRAF6 residue $G_{472}$. The loss of

TRAF6 binding of the LMP1 mutant $V_{380}A$ could be related to a loss of surface contacts between the side chains of LMP1 $V_{380}$ and TRAF6 $V_{474}$.

The most significant difference between LMP1 and the consensus TRAF6-binding sequence, PxExxF/Y/D/E, is that $P_0$ is occupied with glutamic acid in cellular receptors, while LMP1 carries a glutamine at this position. Our experimental data demonstrate that also RANK and CD40 tolerate glutamine at $P_0$ (see Fig. 1h). Thus, comparable effects of Q-E exchange at $P_0$ of all three receptors indicate that there is no significant structural difference between LMP1, RANK and CD40 at this position. This is reflected by our model of LMP1-TRAF6 as compared to the crystal structures of RANK-TRAF6 (PDB 1LB5) and CD40-TRAF6 (PDB 1LB6) (Fig. 4b and Supplementary Fig. 4a to c). It has been shown for cellular receptors that the side chain carboxylate of glutamic acid at $P_0$ forms a strong hydrogen bond network with the backbone amide nitrogen atoms of $L_{457}$ and $A_{458}$[46,60]. According to the model, hydrogen bonds are also formed between $Q_{381}$ at $P_0$ of LMP1 and the backbone of $L_{457}$ and $A_{458}$ of TRAF6 (Fig. 4b). Moreover, a water molecule is observed in the TRAF6-RANK crystal structure, which mediates hydrogen bonds between the carboxylic acid moiety of $E_{346}$ at $P_0$ and $L_{457}$. It is expected that this indirect hydrogen bond is also formed by $Q_{381}$ of LMP1, although the interaction will be weaker because of the neutral character of the terminal amide moiety of $Q_{381}$ as compared to the negatively charged carboxylic acid moiety of $E_{346}$ in RANK. Thus, the different charges of the glutamine and glutamic acid side chains will determine the strength of the electrostatic interactions with TRAF6 at $P_0$.

At $P_1$ to $P_3$, the positioning of the receptor peptides and the interacting TRAF6 residues are comparable between the LMP1-TRAF6 model and the published crystal structure of RANK-TRAF6 (PDB 1LB5). This includes similar orientations of $R_{392}$ of TRAF6 and the tyrosine at $P_3$ of both receptors (Fig. 4c and Supplementary Fig. 4a, c). The $L_{382}A$ mutation at $P_1$ of LMP1 did not impair LMP1-TRAF6 binding (see Fig. 1e). This is consistent with the model, which indicates that hydrogen bonds at this position are formed by the LMP1 backbone with the TRAF6 residues $G_{470}$ and $R_{392}$ and are, hence, invariant to this change of the amino acid side chain of $L_{382}$ (Fig. 4c). Positions $P_1$ and $P_2$ of RANK are occupied with the acidic residues $D_{347}$ and $E_{348}$, respectively, which form additional strong, charged interactions with the basic amine functions of $R_{392}$ and $K_{469}$ of TRAF6[46,60]. The uncharged residues of LMP1 at $P_1$ and $P_2$ are unable to form such interactions and are, in combination with the presence of glutamine at $P_0$, likely the reason for the weaker affinity of LMP1 to TRAF6.

At $P_3$, $Y_{384}$ of LMP1 adopts a stretched orientation, consistent with the one observed for $Y_{349}$ of RANK in the crystal structure PDB 1LB5 (Fig. 4c). This orientation enables the formation of amino-aromatic interactions with $R_{392}$. These interactions explain, together with the $P_1$ and $P_2$ interactions with $R_{392}$, the relevance of $R_{392}$ in TRAF6 binding to LMP1 and RANK. When TRAF6 forms a complex with CD40, $R_{392}$ adopts, according to the CD40-TRAF6 structure PDB 1LB6[46], a distinct conformation, which is not within hydrogen bonding distance to the main chain of CD40. Instead, $F_{238}$ at $P_3$ of CD40 may engage in interactions with $R_{392}$, $F_{410}$ and $V_{474}$ of TRAF6[46,60]. The mutation of $R_{392}$ is likely rescued by the T-shaped aromatic interaction of $F_{238}$ at $P_3$ of CD40 with $F_{410}$ of TRAF6 observed in the crystal structure PDB 1LB6 (see Fig. 2b, compare to 2a and 2c). The observed increase in affinity of LMP1 $Y_{384}F$ towards TRAF6 might be related to the possible formation of hydrophobic interactions with TRAF6, which are not possible with a tyrosine at $P_3$.

Overall, the LMP1-TRAF6 model is consistent with the results of our experimental interaction and mutational studies. The LMP1-TRAF6 complex at CTAR2 resembles features of TRAF6 interaction with RANK but shows differences to the crystal structure of TRAF6 with CD40, the cellular counterpart of LMP1.

## NMR spectroscopy reveals shifting of TRAF6 residues upon LMP1 binding

To further confirm the binding of LMP1 to TRAF6, we recorded NMR spectra of TRAF6 in its free form as well as bound to the LMP1 peptide $G_{378}PVQLSYYD$. Based on a previously published partial backbone chemical shift assignment of TRAF6[61], several TRAF6 residues of interest could be assigned to the signals in the spectra. Of those TRAF6 residues previously tested for their functions in LMP1 binding (see Figs. 2 and 3), $F_{471}$ and $K_{469}$ are highlighted ($R_{392}$ and $Y_{473}$ have not been assigned by Moriya and colleagues[61]) (Fig. 5a). Upon addition of the LMP1 peptide, the signals corresponding to some of the TRAF6 residues were broadened beyond detection, which is indicated by the appearance of the blue resonance of free TRAF6 (Fig. 5a). This indicates a significantly altered chemical environment in the presence of the LMP1 peptide, as expected for "anchor" residues of the protein-protein interaction. The observed line broadening at 5-fold excess of the LMP1 peptide could reflect unusual binding kinetics, or dynamic binding related to conformational dynamics of the protein-protein interface. In any case, a limited set of NMR signals is affected upon complex formation, including shifting of some resonances upon the

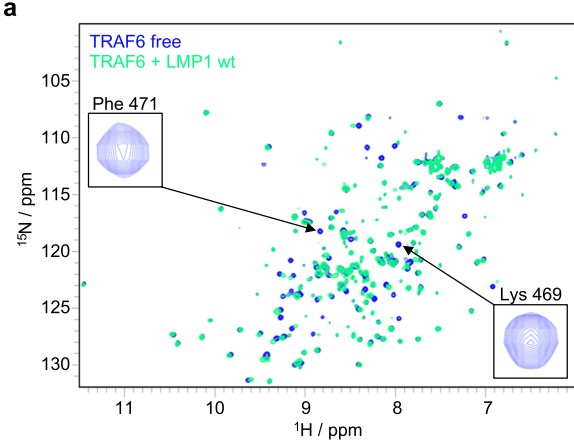

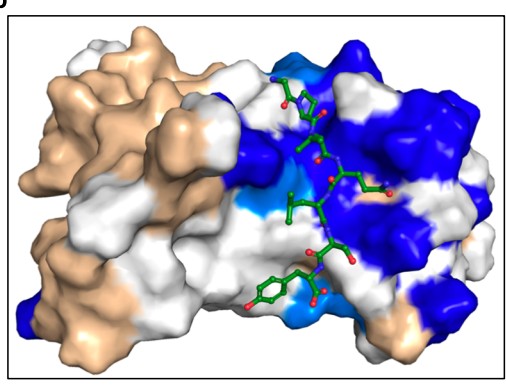

**Fig. 5 | Binding of LMP1 induces intramolecular shifts in TRAF6. a** Overlay of the resonance patterns of the HSQC NMR spectra of TRAF6 alone (blue) and TRAF6 in complex with the LMP1 peptide $G_{378}PVQLSYYD$ (green). Peaks of $F_{471}$ and $K_{469}$ are highlighted. **b** Chemical shifts induced by LMP1 peptide binding mapped at the surface of TRAF6 are highlighted in the in silico LMP1-TRAF6 structure described in Fig. 4. TRAF6 residues showing strong shifts upon LMP1 binding (>0.05) are indicated in dark blue, and residues showing weaker shifts (0.03 − 0.05) in light blue. Residues without a shift upon LMP1 binding are indicated in wheat. TRAF6 residues without an assignment to a specific NMR resonance are shown in light grey. For the latter residues, no NMR-based conclusion is possible regarding their shifting upon LMP1 binding.

addition of the LMP1 peptide, indicating a well-defined binding interface. In summary, the NMR spectra changes detected upon binding of the LMP1 peptide are localized in the close proximity of the predicted LMP1 binding interface and are consistent with our biochemical and molecular modelling data (Fig. 5b).

## LMP1-driven B lymphomas are strictly dependent on TRAF6

CTAR2 provides critical signals for the effective oncogenic transformation of primary B cells by EBV[9,62]. Here, we show that the direct TRAF6 interaction with CTAR2 is required for CTAR2 signalling. To examine if TRAF6 is necessary for the proliferation and survival of LMP1-driven B cell lymphomas, TRAF6 was targeted by an ex vivo CRISPR/Cas9 approach in the two LMP1-dependent B cell lymphomas LMP1-CL 37 and 40 derived from the transgenic *CD19-Cre;R26LMP1^stopfl^:CD3ε^KO^* mouse model[10,63]. The effect of three different gRNAs targeting the gene of interest (GOI) TRAF6 on tumor cell survival was examined. gRNAs targeting LMP1 as positive or the intracellular adhesion molecule 1 (ICAM1) as negative controls were included in parallel transfections. Cell survival was monitored seven days post-transfection as the selection

score of the gRNAs targeting the GOI versus a nontargeting (NT) gRNA directed against an irrelevant Rosa26 sequence (Fig. 6a and Methods).

Knockout of LMP1 resulted in a drastic reduction in the survival of the LMP1-CL 37 and 40 lymphomas (Fig. 6b, c). This result was expected because both lymphomas had been selected for their dependence on LMP1[63] (see Methods). In contrast, targeting ICAM1 did not affect lymphoma survival. More interestingly, we found that the inactivation of TRAF6 by CRISPR/Cas9 caused a massive negative effect on lymphoma survival, which was comparable to the effect of LMP1 targeting itself (Fig. 6b, c). These experiments demonstrate a previously unappreciated critical role of TRAF6 in the survival of LMP1-dependent B lymphoma cells. These findings further suggested that the direct interaction between LMP1 and TRAF6 is an important factor for lymphoma development.

## Disruption of the LMP1-TRAF6 complex interferes with LCL survival

To show that the direct interaction of TRAF6 with CTAR2 contributes essential and sufficient signals to the survival of EBV-transformed B

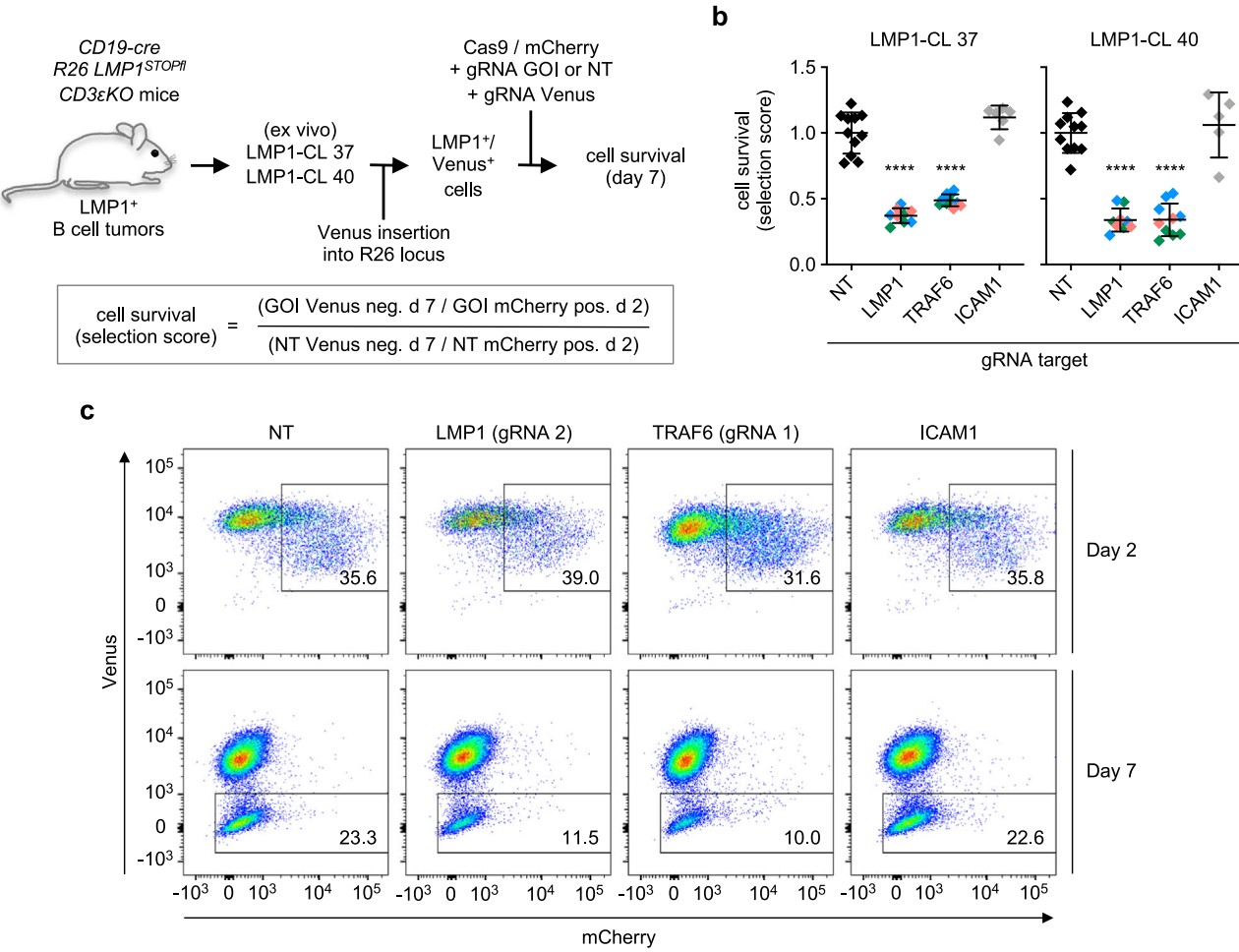

**Fig. 6 | TRAF6 is essential for LMP1-induced B cell lymphoma survival.**
**a** Experimental setup. LMP1-dependent mouse B cell lymphomas LMP1-CL 37 and 40 were derived from *CD19-cre;R26LMP1^STOPfl^:CD3ε^KO^* mice. Venus marker expression was achieved by CRISPR/Cas9-mediated insertion of Venus into the Rosa26 locus. Subsequently, the cells were simultaneously electroporated with two pX330-mCherry-CAS9 vectors, the first construct expressing a gRNA targeting Venus, the second construct expressing either one out of three gRNAs targeting the genes of interest (GOI) LMP1 and TRAF6, or one gRNA targeting ICAM1, or a nontargeting gRNA (NT). Both vectors also expressed mCherry to monitor transfection efficiency. Cell survival was determined by flow cytometry on day 7 post-transfection

as selection score: percentage of Venus-negative GOI gRNA versus NT gRNA cells, normalized for transfection efficiencies (mCherry-positive cells on day 2). **b** The ex vivo knockout of TRAF6 in two LMP1-dependent B cell lymphomas inhibits survival as efficiently as the knockout of LMP1 itself. Data are mean values ± SD of the following number of independent experiments: NT gRNA, *n* = 11 (black); LMP1 gRNA #1, *n* = 3 (green); LMP1 gRNA #2, *n* = 3 (blue); LMP1 gRNA #3, *n* = 3 (salmon); TRAF6 gRNA #1, *n* = 4 (green); TRAF6 gRNA #2, *n* = 4 (blue); TRAF6 gRNA #3, *n* = 2 (salmon); ICAM1 gRNA, CL-37 *n* = 6, CL40 *n* = 5 (gray). Statistics: one-way ANOVA. *p*-values: ****p ≤ 0.0001. Source data and exact p-values in the Source Data file. **c** FACS profiles of one representative experiment with lymphoma LMP1-CL 37.

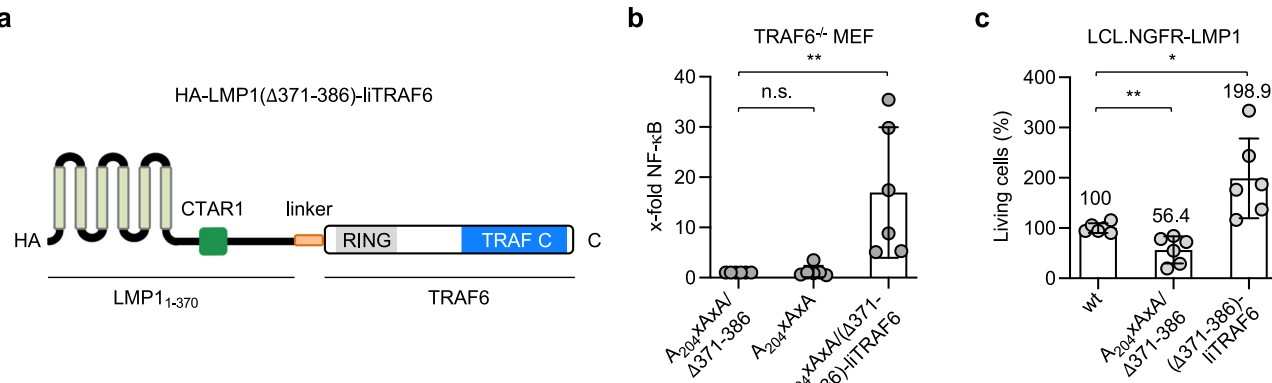

**a** HA-LMP1(Δ371-386)-liTRAF6

**b** TRAF6$^{-/-}$ MEF

**c** LCL.NGFR-LMP1

**Fig. 7 | The direct recruitment of TRAF6 to LMP1 supports LCL survival. a** HA-LMP1(Δ371-386)-liTRAF6 mimics the direct recruitment of TRAF6 to CTAR2. Amino acids 371-386 of CTAR2 were replaced by a flexible linker and TRAF6 wild-type. **b** The forced recruitment of TRAF6 to LMP1 is sufficient to activate NF-κB in the absence of functional CTAR1. TRAF6$^{-/-}$ MEFs were transfected with the indicated HA-LMP1 constructs or HA-LMP1(A$_{204}$xAxA/Δ371-386)-liTRAF6 together with an NF-κB reporter. NF-κB reporter assays were performed. Data are mean values ± SD of six independent experiments. Statistics: one-way ANOVA. **c** LCL.NGFR-LMP1 cells were transfected with doxycycline-inducible pRTS1- mCherry vectors expressing HA-LMP1(Δ371-386)-liTRAF6, HA-LMP1 wild-type or the inactive HA-LMP1(A$_{204}$xAxA/Δ371-386) mutant. Endogenous NGFR-LMP1 activity was silenced by antibody withdrawal and expression of the constructs was induced by the addition of doxycycline. After 16 days, the numbers of DAPI-negative/mCherry$^{high}$ cells were analyzed by flow cytometry. The numbers of living cells expressing HA-LMP1 were set to 100%. Data are mean values ± SD of six independent experiments. Statistics: unpaired T-test, two-tailed. *p*-values: *$p \leq 0.05$, **$p \leq 0.01$, n.s. (not significant). Source data and exact *p*-values in the Source Data file.

cells, we generated an LCL system, in which we were able to switch off endogenous LMP1 activity and, at the same time, induce the expression of a fusion construct of LMP1 lacking CTAR2 and full-length TRAF6, both separated by a flexible linker (Fig. 7a). The resulting construct HA-LMP1(Δ371-386)-liTRAF6 mimics the direct recruitment of TRAF6 to CTAR2 without any further contributions of CTAR2 or factors binding to this domain. In the absence of a functional CTAR1, this fusion protein was able to efficiently activate NF-κB reporter activity in MEFs lacking endogenous TRAF6 (Fig. 7b). All constructs were expressed at similar levels (Supplementary Fig. 5a). This result demonstrates that the forced TRAF6 recruitment to LMP1 is sufficient to replace CTAR2 function. We transfected LCL.NGFR-LMP1 lymphoblastoid cells with doxycycline (dox)-inducible episomal expression vectors for HA-LMP1(Δ371-386)-liTRAF6, as well as HA-LMP1 wild-type or HA-LMP1 lacking functional CTAR1 and CTAR2 as controls (Fig. 7c). An mCherry marker gene, which was driven by the same bidirectional dox-inducible promoter as the LMP1 constructs, indicated expression of the constructs. LCL.NGFR-LMP1 cells depend on the permanent crosslinking of NGFR-LMP1 for their efficient survival and proliferation, because they do not express wild-type LMP1[35]. NGFR-LMP1 activity was switched off in these cells by the deprivation of crosslinking antibodies, and expression of the transfected constructs was induced by dox. After sixteen days, the numbers of living DAPI-/mCherry$^{high}$ cells were analyzed by flow cytometry (Fig. 7c). HA-LMP1(Δ371-386)-liTRAF6 was fully sufficient to maintain the survival of the LCLs in the absence of endogenous LMP1 activity. These results underscored the critical importance of TRAF6 and its direct recruitment to CTAR2 for LMP1 function and LCL survival.

To prove that the direct LMP1-TRAF6 complex can be targeted in vivo, we aimed to inhibit TRAF6 recruitment to LMP1 by peptides to test the effect of LMP1-TRAF6 PPI disruption on LCL survival. Previously, cell-penetrating TRAF6 inhibitory peptides derived from the TRAF6 binding site of RANK were used to inhibit the receptor interaction of TRAF6 as well as RANK signalling[46,64,65]. Because we have shown that LMP1 and RANK bind to the same region at the TRAF6 surface, we reasoned that the RANK-derived peptide should be able to block the TRAF6 interaction with LMP1. A sequence alignment of the TRAF6 inhibitory peptide with LMP1 and CD40 is shown in Fig. 8a. We used a cell-penetrating version of this peptide, fused to the Antennapedia leader sequence, to inhibit the TRAF6 interaction with

LMP1. A peptide comprising the leader sequence alone served as a negative control.

Indeed, the TRAF6 inhibitor peptide blocked the interaction of TRAF6 and LMP1 in AlphaScreen PPI assays with an IC$_{50}$ of 177 nM, while the control peptide was inactive (Fig. 8b, left). TRAF6 binding to LMP1 wild-type and the A$_{379}$xAxxA null mutant demonstrated the dynamic range of the assay and verified that LMP1-TRAF6 inhibition by the peptide was complete (Fig. 8b, right). As expected, the inhibitor peptide did not affect the recruitment of TRAF2 to LMP1 (Fig. 8c). TRAF6 binding to CD40 was also inhibited by the RANK-derived peptide, albeit with reduced efficacy as compared to LMP1 (Fig. 8d).

Finally, we examined the effects of the TRAF6 inhibitor peptide on LCL viability (Fig. 8e). Three lymphoblastoid cell lines, LCL721, HA-LCL3, and LCL.NGFR-LMP1 were incubated for four days in the presence of the TRAF6 inhibitor peptide or the Antennapedia control peptide, respectively. The EBV-negative Burkitt's lymphoma cell line BL41 was included as a negative control. The TRAF6 inhibitor peptide, but not the control peptide, caused a severe reduction in cell viability in the LMP1-dependent LCLs, whereas no such effect was observed in LMP1-independent BL41 cells. This result corroborates our previous results regarding the relevance of TRAF6 for the survival of EBV/LMP1-transformed cells. It further shows that the direct interaction of TRAF6 with LMP1 is essential for LMP1's pro-survival function and might therefore constitute a therapeutic target for inhibitors, such as small molecule LMP1-TRAF6 PPI inhibitors.

## Discussion

With this study, we provide the answer to the long-standing open question of how the signalling-active sequence P$_{379}$VQLSY of LMP1 and the critical signalling mediator TRAF6 are connected at the molecular level. We demonstrate a direct protein-protein interaction of LMP1 and TRAF6 that is the basis for both CTAR2 signalling and the survival of LMP1-transformed B cells. The viral TRAF6 binding motif PVQxxY is unique compared to the known cellular TRAF6 binding motifs in that it carries glutamine instead of glutamic acid at P$_0$. Despite earlier reports suggesting that the exchange of glutamic acid for glutamine at P$_0$ is not tolerated in CD40 without losing affinity towards TRAF6[58], we show here that CD40 and RANK both tolerize this exchange, even though it weakens their affinity towards TRAF6. The lower affinity of TRAF6 towards LMP1, as compared to the cellular receptors CD40 or

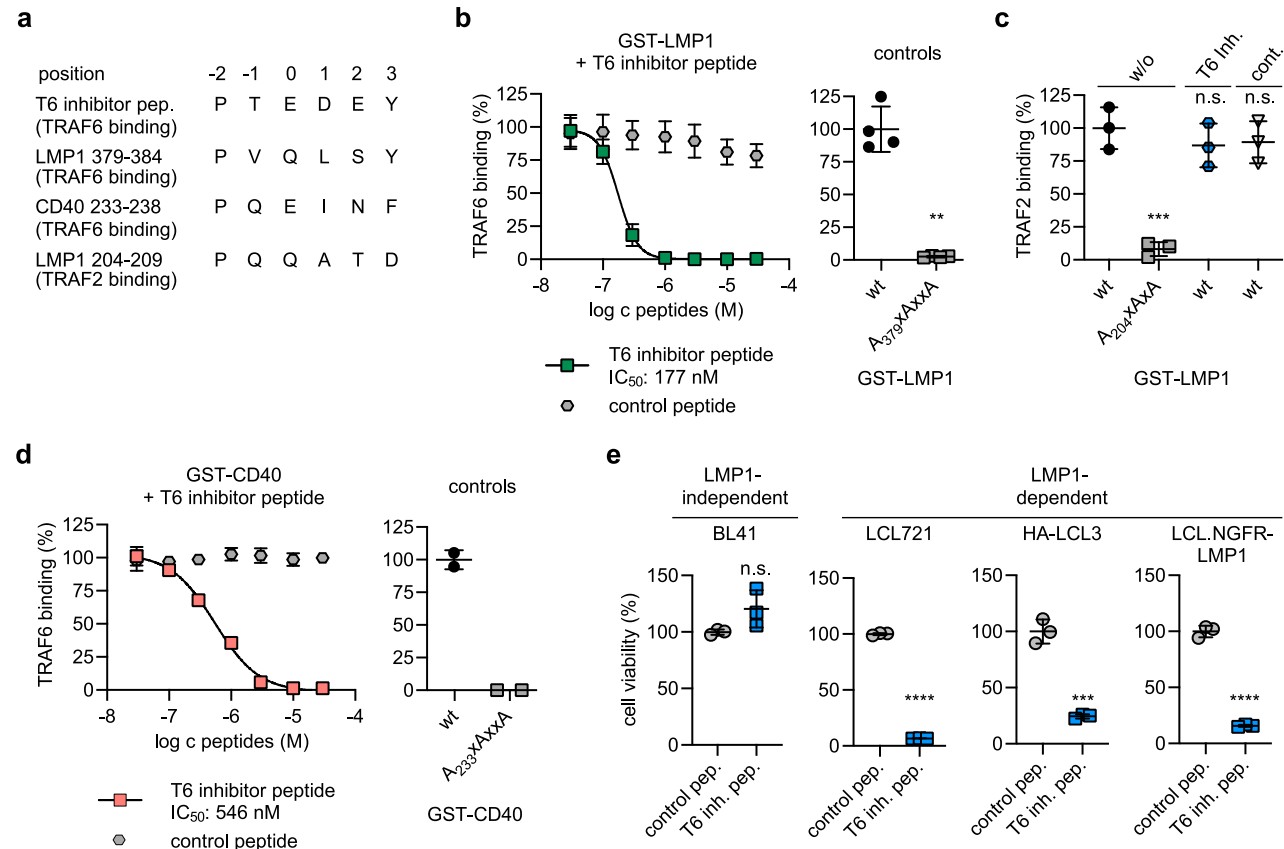

**Fig. 8 | Disruption of the LMP1-TRAF6 complex inhibits the cell survival of EBV-transformed human B cells. a** Alignment of the RANK-derived TRAF6-interacting sequence of the TRAF6 inhibitor peptide DRQIKIWFQNRRMKWKK-RKIPTEDEY with the TRAF6-binding sequences of LMP1, CD40, and the TRAF2-binding sequence of LMP1. **b** TRAF6 binding to LMP1 is efficiently inhibited by the TRAF6 inhibitor peptide. Left, AlphaScreen PPI assay-based dose-response curve of TRAF6 inhibitor peptide (green) with recombinant GST-LMP1$_{181-386}$ and His-TRAF6$_{310-522}$. The Antennapedia leader peptide DRQIKIWFQNRRMKWKK served as a negative control (gray). Right, TRAF6 binding is absent for the LMP1 null mutant A$_{379}$xAxxA. Data are mean values ± SD of four independent experiments. Dose-response curve fitting: 4-parameter fit. Statistics for controls (right): paired T-test, two-tailed. **c** The TRAF6 inhibitor peptide does not affect TRAF2 binding to LMP1. GST-LMP1 and His-TRAF2$_{311-501}$ were incubated in the presence of 30 μM of TRAF6 inhibitor or control peptide in AlphaScreen PPI experiments. Data are mean values ± SD of three independent experiments. Statistics: one-way ANOVA. **d** Peptide-mediated inhibition of TRAF6 binding to CD40 is less effective than that to LMP1. Left, AlphaScreen PPI experiments with GST-CD40 and His-TRAF6$_{310-522}$. Right panel, absent TRAF6 binding to the CD40 null mutant A$_{233}$xAxxA. Data are mean values ± SD of two independent experiments. Dose-response curve fitting: 4-parameter fit. **e** Inhibition of LMP1-dependent cell proliferation of LCLs by the TRAF6 inhibitor peptide. Cells were incubated for four days in the presence of 100 μM of TRAF6 inhibitor peptide or control peptide as indicated. MTT viability assays. The averages of the control replicates were set to 100% viability for each cell line. Data are mean values ± SD of biological triplicates. Statistics: unpaired T-test, two-tailed. *p*-values: *$p ≤ 0.05$, **$p ≤ 0.01$, ***$p ≤ 0.001$, ****$p ≤ 0.0001$, n.s. (not significant). Source data and exact *p*-values in the Source Data file.

RANK, might constitute an important mechanism of balancing the signalling strength of the constitutively active LMP1 within limits that are supportive of cell survival. Highly elevated levels of LMP1 signalling are known to induce cytostasis or even cell death[13].

Our results explain the earlier observation that the mutation of LMP1 Y$_{384}$YD to F$_{384}$FD preserves LMP1's capacity to mediate CTAR2-dependent NF-κB signalling, whereas the mutation to I$_{384}$D abolishes its activity[37]. The Y$_{384}$F exchange at P$_3$ maintains a functional TRAF6 interaction motif, with a decent K$_D$ of TRAF6 interaction (see Fig. 1i). In contrast, the I$_{384}$D mutation is expected to destroy the TRAF6 binding motif. Accordingly, F$_{384}$FD, but not I$_{384}$D, supports B cell transformation[37], which further underscores the relevance of the TRAF6 interaction motif and the direct recruitment of TRAF6 for the transforming capacity of LMP1. Interestingly, the LMP1 F$_{384}$FD mutant is more potent in B cell transformation than the wild-type[37], which reflects the higher affinity of Y$_{384}$F to TRAF6 as compared to wild-type LMP1.

Although the presence of LMP1 and TRAF6 in one signalling complex has been observed previously[20,22,32], direct interaction of both molecules was never demonstrated. In contrast, it was suggested that TRAF6 is recruited to CTAR2 by an indirect mechanism involving TRADD or BS69[20,42]. We now show that not only the direct interaction of TRAF6 with P$_{379}$VQLSY of LMP1 is independent of any further factor but that the forced recruitment of TRAF6 to LMP1 is sufficient to activate NF-κB and to support LCL survival in the absence of a functional CTAR2 sequence, which could recruit further signalling mediators. However, this does not exclude the possibility that TRADD or BS69 act as further stabilizers or modulators of the complex in vivo, dependent on the cellular context or the expression levels of the involved proteins. TRAF6 is critical for both canonical NF-κB and JNK activation by CTAR2[20,28,31,32]. In contrast, TRADD is involved in CTAR2-induced NF-κB but not JNK signalling, whereas BS69 has the opposite function[31,37,38,42]. TRADD and BS69 seem to even compete for LMP1 binding[66,67], which might enable the trimming of TRAF6 signalling towards canonical NF-κB, or noncanonical NF-κB and JNK activation, respectively. But how could TRAF6 and additional factors, such as TRADD and BS69, interact with the same binding sequence at CTAR2 at the same time? TRADD, for instance, requires Y$_{384}$ as a critical residue for its interaction with LMP1[37]. LMP1 oligomerizes to form active signalling complexes[14]. TRAF6 itself can trimerize through its

TRAF domains and further dimerize through its N-terminal RING and Zn finger domains, both together resulting in higher-order oligomerization of TRAF6[46,68]. It is conceivable that LMP1 and TRAF6 form large multimeric network-like complexes at the membrane, in which not all LMP1 molecules must be occupied by TRAF6. This would allow the entry of other factors that interact with $P_{379}$VQLSY. In such higher-order LMP1-TRAF6 patches, TRAF6 and BS69 or TRADD could be recruited to different LMP1 molecules, but still interact with each other to modulate CTAR2 signalling.

We clearly demonstrate that TRAF6 binds to CTAR2 but is unable to interact with CTAR1 directly. However, it has been reported previously that TRAF6 has a role in CTAR1 signalling[20,22,27]. In mouse B cells, TRAF6 coprecipitates with an inducible mCD40-LMP1 fusion protein, an interaction that is dependent on CTAR1 in these cells[22]. The molecular link of TRAF6 to CTAR1 might be related to the ability of TRAF6 to form heterodimers between its own RING domain and the RING domains of TRAF2, TRAF3 and TRAF5[69]. By this mechanism, these TRAF molecules might act as bridging factors for the TRAF6 interaction with CTAR1. Moreover, CTAR1 and CTAR2 cooperate in LMP1 signalling[70]. As shown in the present and in previous work[13], TRAF2, TRAF3 and TRAF5 bind directly to CTAR1. The formation of heterocomplexes between these TRAF molecules bound to CTAR1 and TRAF6 bound to CTAR2 would allow functional interaction of both CTAR domains. This hypothesis is supported by the observation that in mouse B cells both CTARs only achieve robust signalling levels in the presence of TRAF3 and that TRAF3 deficiency completely abrogates the cooperation between CTAR1 and CTAR2[71].

LMP1 expression is a key factor in the pathogenesis of most EBV-associated malignant diseases such as PTLD, HL, DL-BCL, and NPC[1]. There is an urgent medical need for anti-EBV drugs and counteracting LMP1 activity is a promising approach to effectively manage EBV-induced uncontrolled tumor cell proliferation. Pharmacological inhibition of LMP1-induced NF-κB and JNK signalling has already been shown to result in cell death and reduced tumor growth, respectively[24,35,72,73]. However, continuous systemic NF-κB and JNK inhibition might result in adverse side effects that restrict treatment. Instead, the interface between the viral oncoprotein LMP1 and its critical cellular interaction partner offers a more specific target for pharmacological intervention. Here, we identified and validated the direct LMP1-TRAF6 complex as a target for inhibitory molecules, in our case, peptides. Moreover, the deletion of TRAF6 in LMP1-driven mouse B cell lymphomas was as effective as the deletion of LMP1 itself in killing the tumor cells. The next step towards an effective anti-LMP1 drug could be the screening for specific small molecule inhibitors of the LMP1-TRAF6 interaction. Protein-protein interactions are nowadays regarded as well-druggable targets for small molecules[74]. The AlphaScreen-based LMP1-TRAF6 protein-protein interaction assay technology developed in the present work will allow high-throughput screening for such inhibitory small molecules in the future.

## Methods

### Plasmids

The vectors pGEX2T-LMP1$_{181-386}$, pGEX2T-stop, pET17b-His-TRAF2$_{311-501}$, pET17b-His-TRAF6$_{310-522}$[33] pET17b-His-TRAF3$_{375-568}$[75], pGag-pol-IRES-bs$^r$, pEnv-IRES-puro$^r$ [76], pRK5-Flag-TRAF6[33], pSV-NGFR-LMP1[14], pCMV-HA-LMP1(A$_{204}$xAxA) and pCMV-HA-LMP1(A$_{204}$xAxA/Δ371-386)[31], as well as the retroviral vectors pSF91-IRES-GFP-WPRE[77] and pSF91-NGFR-LMP1-IRES-GFP-WPRE[35] have been described. For recombinant human TRAF1 and TRAF5 expression, pET17b-His-TRAF1$_{226-416}$, and pET17b-His-TRAF5$_{363-557}$ were cloned by PCR from cDNAs into the pET17b vector (Novagen). Single amino acid exchanges R392A, K469A, F471A, and Y473A within pET17b-His-TRAF6$_{310-522}$ and pRK5-Flag-TRAF6 were carried out by PCR-based site-directed mutagenesis. The TRAF6 expression vector pET-hSu-TRAF6$_{346-504}$ was cloned by PCR based on His-TRAF6$_{310-522}$. The GST-LMP1 mutants A$_{204}$xAxA, Δ371-

386, A$_{379}$xAxxA, G$_{378}$A, P$_{379}$A, V$_{380}$A, Q$_{381}$A, Q$_{381}$E, L$_{382}$A, S$_{383}$A, Y$_{384}$A, Y$_{384}$D, Y$_{384}$F, and Y$_{384}$G were generated on the basis of pGEX2T-LMP1$_{181-386}$ by PCR. pGEX-2T-CD40$_{216-277}$ and the mutants A$_{233}$xAxxA and E$_{235}$Q were cloned by PCR based on human CD40 cDNA. pGEX-2T-RANK$_{314-355}$ and the corresponding mutant E$_{346}$Q were cloned by PCR based on human RANK cDNA. TRAF6 and the mutants R$_{392}$A, F$_{471}$A and Y$_{473}$A were subcloned into pSF91-IRES-CFP-WPRE. The C-terminal fusion of LMP1Δ371-386 with full-length human TRAF6 wild-type, both separated by a flexible linker with the sequence AGASGGAGASGG, was generated by gene synthesis (Eurofins Genomics). To derive the expression vector pCMV-HA-LMP1(A$_{204}$xAxA/Δ371-386)-liTRAF6, a LMP1Δ371-386-liTRAF6 fragment was subcloned into pCMV-HA-LMP1(A$_{204}$xAxA). The episomal vectors pRTS1-mCherry-HA-LMP1 wild-type, pRTS1-mCherry-HA-LMP1(A$_{204}$xAxA/Δ371-386), and pRTS1-mCherry-HA-LMP1(Δ371-386)-liTRAF6, which carry a bidirectional dox-inducible expression cassette for the respective LMP1 construct together with an mCherry marker gene, were cloned based on the pRTS1 vector[78]. To this end, the gene for the green fluorescence protein of pRTS1 was replaced by mCherry. Subsequently, the LMP1 genes were subcloned into pRTS1-mCherry. Detailed cloning strategies, primer sequences and sequencing results will be made available upon request.

### Cell culture, retroviral transduction, and NGFR-LMP1 activation

Wild-type MEFs and derivatives thereof, TRAF6$^{-/-}$ MEFs[79], and HeLa cells (obtained from the German Collection of Microorganisms and Cell Cultures, GCMC) were kept in Dulbecco's modified Eagle's medium (DMEM, Thermo Fischer, #41966-029). The cell lines HEK293 (obtained from GCMC), HA-LCL3[31], LCL721[80], BL41[81], LCL.NGFR-LMP1[35], and the mouse lymphomas LMP1-CL 37 and 40 were kept in Roswell Park Memorial Institute medium (RPMI 1640, Thermo Fisher, #21875-034). If not indicated otherwise, DMEM and RMPI were supplemented with 10% of fetal bovine serum (FBS; Sigma, #F7524), 1 mM L-glutamine (Thermo Fisher, #25030-024) and antibiotics (50 U per mL penicilline and 50 μg per mL streptomycin). LCL.NGFR-LMP1 cells were additionally supplemented with 100 nM sodium selenite and crosslinking primary and secondary antibodies (see below). All cells were kept at 37 °C in the presence of 5% CO$_2$. Retroviral transduction of MEFs with NGFR-LMP1 and TRAF6 was carried out with the vectors pSF91-NGFR-LMP1-IRES-GFP-WPRE and pSF91-Flag-TRAF6-IRES-CFP-WPRE expressing TRAF6 wild-type or the indicated TRAF6 mutants[35]. For NGFR-LMP1 crosslinking, 1 μg per mL of α-NGFR primary antibody (clone HB8737, ATTC) was added for 1 h to the cells. signalling was activated by further addition of 10 μg per mL of α-mouse IgG/IgM secondary antibody (Dianova, #115-005-068) for the given time points.

### CRISPR/CAS9

The mouse LMP1 B cell tumors LMP1-CL 37 and 40 were derived from two B cell tumors arising in *Rag2*$^{KO}$;*cγ*$^{KO}$ mice inoculated with primary tumor cells from *Rag2*$^{KO}$;*cγ*$^{KO}$ mice reconstituted with fetal liver hematopoietic stem cells and progenitor cells from *CD19-cre*;R26-LMP1$^{STOPfl}$;CD3ε$^{KO}$ animals as described[10]. In these cells, a Venus expression cassette was targeted into the Rosa26 locus by CRISPR/CAS9 as described[82]. Venus-LMP1 B cells were electroporated (Nucleofector, Lonza, human B cell program) with 1 μg of each of two pX330-mcherry-CAS9 vectors[83] encoding varying guide RNAs. In each approach, the first guide RNA (gRNA) targeted Venus while the second gRNA targeted either a nontargeting control sequence not present in the genome, or the gene of interest (GOI). The following gRNA sequences were used: nontargeting (NT) ACTCCAGTCTTTCTAGAAGA, ICAM1 GTTTGAGCTGAGCGAGATCG, LMP1 #1 TTAATCTGGATGTATTACCA, LMP1 #2 CCAAAACAGTAGCGCCAAG, LMP1 #3 AATCATCGGTAGCTTGTTG, TRAF6 #1 TGTGGAGTTTGACCCACCTC, TRAF6 #2 TCTGGACGACATCCCCGGGA, TRAF6 #3 CATCGCACGGACGCAAAGCA.

The electroporation efficiency, measured as the frequency of mCherry-positive cells on day two after electroporation, was determined by flow cytometry with a Becton Dickinson LSRFortessa flow cytometer. On day seven, the loss of Venus was measured by flow cytometry as a surrogate marker for the number of surviving targeted cells. In the case of a lethal coexpressed GOI gRNA, Venus-negative cells will not survive, and their relative numbers are reduced. The given selection score is defined as the frequency of Venus-negative cells on day seven over the frequency of mCherry-positive cells on day two, normalized to the ratio observed with the nontargeting control gRNA. FACS data were analyzed by FlowJo software. The gating strategy is exemplified in Supplementary Fig. 5b. Animal housing and experiments to generate LMP1 lymphomas were approved by the Landesamt für Gesundheit und Soziales Berlin (G0049/15, G0374/13, and G0135/11). Mice were kept in groups in individually ventilated cages with a 12 h dark-light cycle at $22 \pm 2\,°C$ and $55 \pm 5\%$ relative humidity. They had access to food and water ad libitum, as well as to nest building material and shelter.

### Electroporation of LCL.NGFR-LMP1 cells
LCL.NGFR-LMP1 cells were cultured in RPMI full medium containing crosslinking antibodies (see above). The cells were deprived of antibodies one week before electroporation to downregulate NGFR-LMP1 activity. LCL.NGFR-LMP1 cells ($1 \times 10^7$) were electroporated in a Bio-Rad Gene Pulser II at 180 V and 975 µF with 5 µg of pRTS1-mCherry-HA-LMP1 wild-type, pRTS1-mCherry-HA-LMP1($A_{204}$xAxA/$\Delta$371-386) double mutant, or pRTS1-mCherry-HA-LMP1($\Delta$371-386)-liTRAF6 together with 10 µg of salmon sperm DNA. After electroporation, the cells were cultured in the absence of crosslinking antibodies and in the presence of 1 µg per mL doxycycline to induce the expression of the LMP1 constructs and mCherry. On day 16 post electroporation, the cells were washed with PBS containing 5% FBS, stained with DAPI (4′,6-diamidino-2-phenyindole) and analyzed by flow cytometry in a Becton Dickinson LSRFortessa flow cytometer. Recorded data were processed with FlowJo 10 software. Numbers of DAPI-negative (living)/mCherry$^{high}$ lymphocytes were expressed as percentages versus the average of all parallel independent LMP1 wild-type samples, which was set to 100%. The gating strategy is exemplified in Supplementary Fig. 5c.

### Purification of His-tagged proteins
For protein expression, 400 mL of bacterial cultures (BL21 Codon Plus RIPL, Agilent Technologies, #230280) in lysogeny broth (LB)-medium (supplemented with 50 µg per mL Amp and 34 µg per mL Cam) were induced with 0.1 mM IPTG at an $OD_{600}$ of 0.8–1.0. Protein expression was carried out overnight at 20 °C while shaking at 200 rpm. Cells were pelleted (3500 x $g$, 20 minutes, 4 °C), resuspended in sodium phosphate buffer (50 mM sodium phosphate, 300 mM NaCl, 10 mM imidazole, pH 8.0) supplemented with a protease inhibitor cocktail (cOmplete, Roche; following the manufacturer's instructions) and lysed by adding 1 mg per mL (final concentration, f. c.) lysozyme (Merck, #9001-63-2) followed by sonication on ice (three times 10 seconds at 30–50% amplitude, on ice). Lysates were incubated on ice for 20 minutes and cleared from insoluble debris by centrifugation (10,000 x $g$, 30 min, 4 °C), loaded onto Ni$^{2+}$-NTA agarose beads (Qiagen, #30210), and incubated for 1 h at 4 °C. The beads were washed stepwise with 20 mM, 50 mM and 100 mM imidazole in sodium phosphate buffer. His-TRAF proteins were eluted with 500 mM imidazole in sodium phosphate buffer. For buffer exchange, proteins were loaded onto DextraSec Pro10 columns (AppliChem) and eluted in PBS according to the manufacturer's instructions. Samples were supplemented with 10% glycerol (f. c.) and stored at −20 °C.

### Purification of GST-tagged proteins
For protein expression, 400 mL bacterial cultures (DH5α or BL21 Codon Plus RIPL) were transformed with the corresponding expression plasmids, grown in LB medium supplemented with appropriate antibiotics and induced with 0.1 mM IPTG at $OD_{600}$ = 0.8–1.0. Protein expression was carried out overnight at 20 °C. Cells were pelleted (3500 x $g$, 20 minutes, 4 °C), resuspended in PBS (pH 7,4) supplemented with 0.1% Tween-20 and protease inhibitor cocktail (cOmplete, Roche), and lysed by adding of 1 mg per mL (f. c.) lysozyme followed by sonication. Lysates were incubated on ice for 20 minutes, cleared from insoluble debris by centrifugation (10,000 x $g$, 30 minutes, 4 °C), loaded onto glutathione Sepharose 4B beads (GE Healthcare, #17-0756-01) and incubated for 2–4 h at 4 °C. Beads were washed three times with PBS, and bound proteins were eluted with 500 mM reduced glutathione in PBS. For buffer exchange, all proteins were loaded onto DextraSec Pro10 columns (AppliChem) and eluted in PBS according to the manufacturer's instructions. Samples were supplemented with 10% glycerol and stored at −20 °C.

### AlphaScreen PPI assays
AlphaScreen experiments were performed in 96- or 384-well plates (Perkin Elmer, OptiPlate). In both assay formats, His-TRAF and GST-LMP1, GST-CD40, or GST-RANK proteins were incubated together in 40 µL of PBS (protein concentrations were calculated to the final reaction volume of 60 µL) supplemented with 0.5% BSA (Sigma, #A7030) and 0.1% Tween-20 (AppliChem, #A4974), for 1 h at room temperature (RT). If not indicated otherwise, standard protein concentrations were 300 nM for His-TRAF proteins and 100 nM for GST-LMP1 proteins. Ni-NTA-acceptor (Perkin Elmer, #6760619 C) and GST-acceptor (Perkin Elmer, # 6765300) beads were added each to a final concentration of 1–4 µg per mL in a final reaction volume of 60 µL and incubated for 1 h at RT in the dark. AlphaScreen PPI signals were measured in a CLARIOstar reader (BMG Labtech GmbH).

### Pulldown experiments
Ten microliters of a 50% slurry of glutathione Sepharose 4B beads (GE Healthcare) was loaded with 80 µg of purified GST-LMP1 or 40 µg of GST, as indicated, for 2–4 h at 4 °C in 500 µL PBS in an overhead shaker. The beads were pelleted at 500 x g for 5 minutes at 4 °C, and washed twice with PBS. Beads loaded with GST-LMP1 or GST control were then further incubated with 1,5 µg of His-TRAF proteins in 500 µL PBS supplemented with 0.1% BSA and 0.1% Tween-20 for 1 h at 4 °C. The beads were pelleted, washed three times with PBS containing 0.1% Tween and resuspended in 75 µL of Laemmli SDS sample buffer. Samples were loaded onto 15% SDS-PAA gels and proteins were separated by SDS-PAGE. GST-LMP1 proteins were visualized by incubation of the SDS-PAA gels in 20% acetic acid containing 1% Coomassie Blue G-250 (Serva). Free dye was removed by repeated incubation in 10% acetic acid. The amounts of His-TRAF proteins were analyzed by immunoblotting using the α−6*His-tag antibody.

### Immunoblotting
Immunoblotting was essentially performed as described[35]. Proteins were detected on nitrocellulose membranes (Bio-Rad, #1620115) using the following primary antibodies: α−6*His-tag (clone 4A4 or 2F12, source: antibody facility of the Helmholtz Center Munich, hybridoma supernatants were used at a 1:10 dilution)[75], α-TRAF6 (H-274, Santa Cruz Biotech., #sc-7221, used at 1:1000), α-HA-tag (12CA5, Sigma-Aldrich, #11583816001, used at 0.4 µg per mL), α-IκBα (C-21, Santa Cruz Biotech., #sc-371, used at 1:500) and α-Tubulin (B-5-1-2, Santa Cruz Biotech., #sc-23948, used at 1:500). Horseradish peroxidase-coupled antibodies were used as secondary antibodies (Cell Signalling Technology, #7074 or #7076, respectively, both used at 1:5000). ECL

signals were captured on X-ray films (Agfa Healthcare) and quantified by densitometry using the ImageJ software. Uncropped scans of immunoblots indicating molecular weights are provided in Supplementary Fig. 6.

## Immunoprecipitation

HEK293 cells were transfected with 1 μg of pRK5-Flag TRAF6 plasmids, 1 μg of pCMV-HA-LMP1 and 6 μg of pRK5 per 10 cm cell culture dish using PolyFect Transfection Reagent (Qiagen, #301107) in RPMI without supplements. Cells were incubated with the plasmid transfection mix for 4 h before the medium was changed to RPMI full medium. After 24 h, the cells were lysed in NP40-lysis buffer (50 mM HEPES pH 7.5, 150 mM NaCl, 5 mM EDTA, 0.5 mM sodium orthovanadate, 0.5 mM PMSF, 0.5 mM sodium molybdate, and cOmplete protease inhibitor cocktail). Lysates were cleared by centrifugation at 15000 x g for 10 min at 4 °C, and the protein concentration was adjusted to 1 mg per mL. For immunoprecipitation 4 mL of lysate was incubated with the α-Flag antibody 6F7 (Sigma-Aldrich, #F3165; covalently coupled to protein-G-Sepharose) for 1 h at 4 °C. The beads were pelleted at 15000 x $g$ for 30 seconds at 4 °C, washed twice with NP40-lysis buffer, and analyzed by immunoblotting.

## NF-κB reporter assay

TRAF6$^{-/-}$ cells ($6 \times 10^4$) were seeded per well of a 6-well plate in DMEM full medium the day before transfection. Cells were transfected with 2 μg of the indicated pCMV-HA-LMP1 plasmids together with 0.5 μg of pRK5-Flag-TRAF6 plasmids or empty vector, 0.05 μg of NF-κB luciferase reporter 3xκB-Luc and 0.2 μg of pPGK-Renilla housekeeping control reporter using the PolyFect transfection reagent (Qiagen)[33]. After 4 h the transfection mix was removed, and the cells were kept overnight in DMEM full medium. NF-κB reporter and Renilla control activities were measured with the Dual-Luciferase reporter assay kit (Promega, #E1910) following the manufacturer's instructions. Samples of each lysate were analyzed for LMP1 and TRAF6 expression by immunoblotting.

## Peptide arrays

Arrays were synthesized on a Multipep Synthesizer (Intavis Bioanalytical Instruments) on derivatized cellulose (amino-Peg500 UC540, acid-hardened, loading 400 nmol per cm², Intavis). After peptides were spot-assembled and deprotected, membranes were washed extensively with dichloromethane, N-methylpyrrolidone, and ethanol and stored at −20 °C until use. Prior to use, filters were rinsed for 10 sec in ethanol and washed three times for ten min each at 50 rpm in PBST (PBS, 0.1% Tween-20) under gentle rocking. Filters were blocked by incubation in PBST supplemented with 5% (w/v) nonfat dry milk powder for 2 h at RT and three washing steps in PBST for 20 min. Subsequently, nonspecific antibody binding sites were blocked by incubation of the filters at 4 °C overnight with either TRAF6 (goat α-TRAF6 C-20 #sc-6223, Santa Cruz Biotech.) or TRAF2 primary antibody (rabbit α-TRAF2 C20, #sc-876, Santa Cruz Biotech.), diluted at 1:1000 in PBS supplemented with 5% (w/v) nonfat dry milk powder. Following three washing steps with PBS, the filters were incubated with horseradish peroxidase-conjugated α-goat IgG (Dianova, #305-035-003) or α-rabbit IgG (Cell signalling Technology, #7074S) secondary antibody, diluted 1:5000 in PBS supplemented with 5% (w/v) nonfat dry milk powder. Filters were developed with ECL and no nonspecific antibody binding was detected. To detect TRAF6 or TRAF2 binding to the immobilized peptides, filters were washed with PBST and subsequently incubated with 10 μg per mL recombinant purified His-TRAF6 or His-TRAF2 protein in PBST supplemented with 5% (w/v) nonfat dry milk powder for 4 h at RT. After washing the filters in PBST, TRAF protein binding was analyzed by incubation of the filters with TRAF6 (C-20, #sc-6223, Santa Cruz Biotech., used at 1:1000) or TRAF2 (C-20, #sc-876, Santa Cruz Biotech., used at 1:1000) primary and the respective

Horseradish peroxidase-coupled secondary antibodies (Dianova, #305-035-003, used at 1:500, or Cell Signalling Technology, #7074, used at 1:5000). ECL signals were captured on X-ray films (Agfa Healthcare).

## Structural modelling

A structural model of the LMP1 peptide G$_{378}$PVQLSY interacting with TRAF6 was derived with the Schrödinger platform for drug discovery (Schrödinger platform release 2021-3 of Schrödinger, Inc., 1540 Broadway, New York, NY 10036), from the structure of a RANK-TRAF6 complex (PDB 1LB5)[46]. First, the structure was prepared with the Protein Preparation Wizard and associated modules (all part of the Schrödinger platform) using default settings. This process included the assignment of bond orders, addition of hydrogens, generation of het states (with Epik), addition of missing amino acid side chains (with Prime), H-bond assignment by sampling water orientations, assignment of charge states at physiological pH (with PROPKA), and restrained optimization (with the OPLS4 force field[84], to converge heavy atoms to an RMSD of 0.30 Å). Mutations to transform RANK into LMP1 were performed manually with the "mutation" feature available via the context menu within Maestro (the graphical user interface of the Schrödinger platform). The LMP1-TRAF6 model was then minimized with Prime, again using the OPLS4 force field with default settings. Images were generated with PyMol Molecular Graphics System (Version 2.4.2, Schrödinger, Inc.).

## NMR Spectroscopy

For NMR, a His-SUMO-TRAF6 fusion construct was expressed in *E. coli* Rosetta2 DE3 cells from pET-hSu-TRAF6$_{346-504}$. The recombinant protein was uniformly labeled with $^{15}$N by growing expression cultures in $^{15}$N-autoinduction medium[85]. TRAF6 protein was purified from cell lysate via IMAC using a Ni-NTA column. After elution, the protein was transferred to 100 mM Tris-HCl pH 8.0, 300 mM NaCl, 20 mM imidazole, and 5 mM β-mercaptoethanol. SUMO-hydrolase dtUD1 was added (ratio 1:50) and incubated overnight at 4 °C. After cleavage, the protein was subjected to a second Ni-NTA affinity chromatography step and then further purified via size-exclusion chromatography using a Superdex 75 10/300 GL column (Äkta system, GE Healthcare). SEC buffer was PBS (37 mM NaCl, 2.7 mM KCl, 10 mM Na$_2$HPO$_4$, 1.8 mM KH$_2$PO$_4$ pH 7.4, 5 mM β-ME) and was subsequently used in the following NMR experiments. NMR spectra were recorded at 298 K using a Bruker Avance 600 spectrometer with a QCI cryogenic probe and topspin v.3.2 software (Bruker BioSpin). Spectra were processed using NMRDraw v.8.7 of NMRPipe software[86] and analyzed with the CCPN Analysis software v.2.4.1[87]. NMR titrations were performed by recording $^1$H,$^{15}$N HSQC experiments. For titrations of TRAF6, samples of 90 μM $^{15}$N-labeled TRAF6 in PBS pH 7.4, 10% D$_2$O, and 5 mM β-ME were used. Samples contained either no ligand (reference), or a five-fold excess of unlabeled LMP1 peptides (PSL) as indicated. Shifts were highlighted in the TRAF6-LMP1 model using the PyMol v2.4.1 software (Schrödinger LLC). TRAF6 residues were annotated based on the previously published partial backbone chemical shift assignment of TRAF6[61].

## Confocal immunofluorescence

HeLa cells were seeded onto cover glasses (Thermo Scientific, Menzel) in 24-well plates one day before transfection. Cells were transfected at 50−60% confluency with 50 ng of pCMV-HA-LMP1 plasmids and 100 ng of pRK5-Flag-TRAF6 plasmids, adjusted to a total of 500 ng DNA per sample with pRK5 empty vector, using Lipofectamine 2000 (Thermo Fisher Scientific) according to the manufacturer's instructions. At 24 h posttransfection, cells were washed once with PBS and fixed with 4% paraformaldehyde (Sigma-Aldrich) for 20 min at RT, followed by permeabilization with 0.3% (v/v) Triton X-100 and blocking with 5% (v/v) FBS in PBS

for 1 h at RT. For immunostaining, the cells were incubated overnight with the primary antibodies α-HA (1:1000, clone 16B12, BioLegend) and α-DYKDDDDK (1:1000, clone L5, BioLegend) at 4 °C. The secondary antibodies goat α-mouse Alexa 488 (1:500, cross-adsorbed, Invitrogen, #A-11029) and goat α-rat Alexa 555 (1:500, cross-adsorbed, Invitrogen, #A-21434) were added for 1.5 h at RT. Microscope slides were sealed with cover glasses using Mowiol mounting medium (Sigma-Aldrich). Nuclei were stained with Hoechst 33258 (Sigma-Aldrich). All samples were imaged using a Leica TCS SP5 confocal microscope with 405 nm, 488 nm, 543 nm or 633 nm laser lines, scanning each channel separately under image capture conditions that eliminated channel overlap. For each single experiment, 10 transfected cells positive for both constructs were randomly captured as single-cell images for quantification. The resulting images were then analyzed using the Leica TCS SP5 colocalization analyses application (LAS AF Version 2.7.3.9723). Analysis was performed with default parameters.

## MTT cell viability assay

At day zero, $10^4$ cells were seeded in 96-well plates in the presence of 100 μM TRAF6 inhibitor peptide (NBP2-26506, Novus Biologicals) or control peptide in 50 μL of RPMI full medium. The cells were incubated at 37 °C and 5% $CO_2$. On day four, 10 μL of 0.5 mg per mL MTT (3-(4,5-dimethylthiazol-2-yl)−2,5-diphenyltetrazolium bromide) was added. After 4 h, MTT turnover was analyzed after adding 200 μL of HCl:isopropanol (ratio 1:24). The blue reaction product was measured at 550 nm with a reference filter at 690 nm.

## Statistics

All statistical analyses were performed with Prism v10.1.0 software (GraphPad Software Inc.). Two-way ANOVA was performed on experiments with two factors of variance (Figs. 1e and 3a). Ordinary one-way ANOVA with multiple comparisons was performed on experiments with one factor of variance (Figs. 2a–e, 3c, 6b, 7b, 8c and Supplementary Fig. 2a). For $K_D$ determination, the data from each experiment were cleaned for the hook effect (signal depletion by oversaturation of the beads) and analyzed using the equation one site-specific binding with hill slope (Figs. 1f–i, and Supplementary Fig. 1f). The $IC_{50}$ was determined using the log(inhibitor) vs. response – variable slope (four parameters) equation (Fig. 8b and d). A paired t-test was used for matched samples (Fig. 8b) and an unpaired t-test was used for unpaired samples (Figs. 7c and 8e).

## Reporting summary

Further information on research design is available in the Nature Portfolio Reporting Summary linked to this article.

# Data availability

The data supporting the findings of this study are available from the corresponding authors upon request. Unique cell lines are available upon request. Detailed molecular cloning strategies and a PDB file of the LMP1-TRAF6 model are available upon request. The following PDB files were used for modeling and structural analysis and are available in public databases: 1LB5 and 1LB6. Uncropped images of immunoblots are provided in the Supplementary Information. Source data are provided with this paper.

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

## Acknowledgements

We thank Ichio Shimada of the University of Tokyo for generously sharing the NMR assignment data of TRAF6. We are grateful to Elisabeth Kremmer for His-tag antibodies and Michael Sattler for helpful discussion. This work was supported by German Research Foundation (DFG) grants Ki 825/1-3 to A.K. and grant RTG 2467 (project number 391498659) to S.M.F., grant TTU 07.710 from the German Research Center for Infection Research (DZIF) to A.M., and by grants TTU 07.802, TTU 07.809, and TTU 07.825 from the German Research Center for Infection Research (DZIF), and the Life Science Foundation to A.K.

## Author contributions

A.K. designed and supervised the project, analyzed and interpreted data, and wrote the manuscript, F.G. performed experimental studies, analyzed and interpreted data, and wrote the manuscript, M.O., T.S., D.W., K.R.S, and A.M. performed experimental studies, analyzed and interpreted data, H.K. and A.G. performed experimental studies, S.M.F. contributed essential material and protocols, B.B. and G.P. analyzed and interpreted data, J.K. conducted the molecular modelling, analyzed and interpreted data. All authors reviewed the manuscript.

## Funding

## Competing interests

The authors declare no competing interests.
