## [Peer Review File · Nature Communications]

Epstein-Barr virus-driven B cell lymphoma mediated by a direct LMP1-TRAF6 complexREVIEWER COMMENTS

Reviewer #1 (Remarks to the Author):

The manuscript by Giehler et al. demonstrated that TRAF6 interacts directly with a TRAF6 binding motif within CTAR2, and that recruitment of TRAF6 to LMP1 through the motif is essential for NF- κ B activation and survival of LMP1-driven B cell lymphoma. The experimental results at the molecular level are sound; however, as discussed below, the structural model is somewhat prejudiced, and its uniqueness as the interaction mode seems to be exaggerated. The reviewer suggests revising the manuscript to discuss the complex model more carefully and support their claims with a few additional experiments. Below is the list of concerns.

1. The CTAR1 sequence, P204xQxT, is similar to the CTAR2 sequence, P379VQLSY384; however, only CTAR 2 can bind to TRAF6, why? The author might want to show that the Y384D mutation abolishes the TRAF6 interaction.
2. In figure 3, TRAF6 wildtype, but none of the binding-defective mutants, was able to rescue CTAR2 signaling to NF- κ B. How about LAMP1 A204xAx/Q381E and A204xAx/T384F mutants, which showed enhanced affinity to TRAF6?
3. The author claimed that "the TRAF6 mutant R392A revealed a striking difference regarding LMP1 and CD40 binding". However, it is not so striking if LMP1 is compared with RANK. It is already shown that RANK interaction was not impaired by the TRAF6 mutant R392A (Nature volume 418, pages443–447 (2002)). The finding is rather exaggerated here.
4. The molecular model of the LMP1-TRAF6 complex is proposed without any explanation of the starting complex or quality of the resultant model. Why did the author choose the MAVS-TRAF6 complex that is introduced rather abruptly? Wouldn't it be better to start from the CD40-TRAF6 complex (1LB6), especially when the authors compare that structure of the LMP1-TRAF6 complex with the CD40-TRAF6 complex? In addition, how was energy minimization conducted, and how was the quality of the minimized structure assessed?
5. The author claims that "the unique conformation of the LMP1-TRAF6 interface at P1 to P3 enables additional stabilizing contacts between LMP1 and TRAF6, which are not present in CD40-TRAF6, and may well explain why LMP1 tolerates glutamine at P0." The interaction is not unique to the LMP1-TRAF6 interface, and a similar interaction was observed in the RANK-TRAF6 interaction as well as in the MAVS-TRAF6 interaction. One easy experiment to support the claim for glutamine toleration is to see if the interaction between RANK and TRAF6 tolerates the E to Q mutation. Furthermore, the structural model cannot explain why the LMP1 Y384F mutation improved the affinity. From the author's claim, the LMP1 Y384F mutation disrupts the P1-P3 interface and makes the protein would not allow to have Q in the P0 position and thus cannot bind. These considerations indicate that the structural model is rather questionable and is prejudiced.
6. The author claims that "Upon addition of the LMP1 peptide, the peaks corresponding to these residues are broadened beyond detection, indicating that these residues contribute strongly to binding." This is a misleading statement. The signal broadening would not indicate the contribution to the binding. It indicates that these residues are within the region affected by the interaction. In addition, the signal broadening indicates the presence of multiple conformations in the complex, which poses a question to the correctness of discussing with a single static conformation from the binding model.
7. The CSP mapping poorly matches the proposed model as it extends horizontally in the figure. Especially, the chemical shift mapping does not show any perturbation near the P3 site in the proposed model, while the residue strongly contributes to the interaction. The author might want to

indicate the residues not assigned in the figure.

Reviewer #2 (Remarks to the Author):

Giehler et al. describe much work to establish that TRAF6 binds the CTAR2 directly and thereby resolve a subject of uncertainty in the mechanism by which the LMP1 oncogene signals via TRAF6 in cells. They wish also to test the role of this binding in cell survival. They show that targeting both LMP1 and TRAF6 in two murine cell lines with CRISPR and gRNAs decreases their survival and acknowledge that this finding does not establish a role for the direct binding of TRAF 6 and LMP1 in the survival of these cells. They use cell permeable peptides to block this binding, somewhat unconvincingly to test their contention directly. For example, the data for the cell line HA-LCL3 in Fig. 7E is either misleading or indicates that 90% of the cells are killed by the control peptide. The authors need to provide some other approach to demonstrate a need for direct binding for the survival of these cells. For example, can they express CTAR2-null LMP1 with a coiled-coil domain and TRAF6 with its partner coiled-coil domain so that they can only interact with each other (see: Protein Sci. 2012 Apr; 21(4): 511-519) to support LMP1's functions? Some positive test for a role for direct binding in cell survival is needed to establish the authors contention.

Reviewer #3 (Remarks to the Author):

Giehler at all provided new insight of LMP1-TRAF6 signaling. The authors showed that direct interaction of TRAF6 to LMP1. They analyzed the interaction site of TRAF6 on the intercellular part of LMP1. Finally, they showed that the direct interaction of TRAF6 to LMP1 is critical for NF-kB activation and survival of LMP1-driven B cell lymphoma.

Although the current study contains interesting points in the field of TRAF biology, the direct interaction of TRAF6 and LMP1 is not novel finding. The function of LMP1-TRAF6 was already studied at 2012 with several studies and the direct interaction of LMP1 to TRAF6 was also analyzed at 2011.

1. Molecular mechanisms of TNFR-associated factor 6 (TRAF6) utilization by the oncogenic viral mimic of CD40, latent membrane protein 1 (LMP1) J Biol Chem 2011 Mar 25;286(12):9948-55
2. TRAF6 is a critical regulator of LMP1 functions in vivo Int Immunol . 2014 Mar;26(3):149-58.
3. TRAF binding is required for a distinct subset of in vivo B cell functions of the oncoprotein LMP1 J Immunol . 2012 Dec 1;189(11):5165-70. doi: 10.4049/jimmunol.1201821.

The evidence of direct interaction between LMP1 (suggested peptide) and TRAF6 is still vague. Suggest to perform SPR and ITC (peptide/protein interaction).

The complex structure modeling was not perfect. the validation should be done. Structure modeling (peptide/protein) contains limited information.

General response to the reviewers:

We greatly appreciate the thoughtful comments and criticisms of our peer reviewers, which helped us to further improve our manuscript. In response to, and stimulated by their criticisms we added the following new data and experiments:

1. We generated a new and coherent AlphaScreen TRAF6 protein-protein-interaction (PPI) dataset (new Figures 1f to 1i, Figure 2c, Supplementary Figures 1d and 1f), which now includes also RANK as a third receptor molecule, besides of LMP1 and CD40. Furthermore, we added new point mutants to address the questions of reviewer 1 (for details see below). Of note, K_D values of the first version of the manuscript have been reproduced.
2. Stimulated by the comments of reviewers 1 and 3 and by our new RANK-TRAF6 interaction data, we repeated the molecular modeling of the LMP1-TRAF6 complex, now starting from the RANK-TRAF6 crystal structure PDB 1LB5. The resulting LMP1-TRAF6 model is almost identical to the model shown in the first version of the manuscript, which was derived from the MAVS-TRAF6 crystal structure PDB 4Z8M. Figure 4 was fully replaced by the new model. In light of our new AlphaScreen PPI interaction data and mutant analyses we substantially revised the discussion of the LMP1-TRAF6 model. We are convinced that the new interpretation of the model is sound, explains our experimental data and is also in line with previously published results.
3. To be able to firmly establish a role for the direct binding of TRAF6 and LMP1 in the survival of EBV-transformed human B cells, we designed and established a new experimental system, in which the direct recruitment of TRAF6 to LMP1 via CTAR2 is mimicked by the C-terminal fusion of LMP1 lacking its CTAR2 domain to wild-type TRAF6 (new Figures 7a to 7c). This construct maintained the survival of lymphoblastoid cells in the absence of LMP1 wild-type activity, which was shut off by the deprivation of LCL.NGFR-LMP1 cells from crosslinking antibodies. For more details see below in our response to reviewer 2 and in the manuscript itself.
4. The experiments testing the effects of a TRAF6 inhibitory peptide, which blocks the LMP1-TRAF6 interaction, were repeated with new batches of peptides and one more LCL, LCL.NGFR-LMP1 (new Figure 7h) with essentially the same results as in the previous Figure 7E. The presentation of the data was modified to make the data more easily comprehensible. For more details see our response to reviewer 2.

One error in Figure 7g was corrected (effect of the TRAF6 inhibitory peptide on CD40-TRAF6, previous Figure 7D) that came to our attention during the repeated inspection of the data processing of our primary data. During the transfer of the primary AlphaScreen data from Excel into Prism, the peptide concentrations of this single experiment had been shifted by mistake. We corrected this error, resulting in the new graph shown in Figure 7g (left) and the new IC_{50} of 546 nM. The conclusions from this experiment are not affected by this correction.

Major changes in the text have been highlighted in red. Minor changes including the improvement of wording were not highlighted for better clarity.

Specific response to the reviewers:

Reviewer #1 (Remarks to the Author):

The manuscript by Giehler et al. demonstrated that TRAF6 interacts directly with a TRAF6 binding motif within CTAR2, and that recruitment of TRAF6 to LMP1 through the motif is essential for NF- κ B activation and survival of LMP1-driven B cell lymphoma. The experimental results at the molecular level are sound; however, as discussed below, the structural model is somewhat prejudiced, and its uniqueness as the interaction mode seems to be exaggerated. The reviewer suggests revising the manuscript to discuss the complex model more carefully and support their claims with a few additional experiments. Below is the list of concerns.

1. The CTAR1 sequence, P204xQxT, is similar to the CTAR2 sequence, P379VQLSY384; however, only CTAR 2 can bind to TRAF6, why? The author might want to show that the Y384D mutation abolishes the TRAF6 interaction.

Response: It is, in fact, an interesting question why CTAR1 does not interact with TRAF6 directly (see Figures 1b, 1c, 1e, and 1f), although it harbors the extended TRAF6 interaction consensus motif PxQ/ExxF/Y/D/E, which emerged from our studies on TRAF6 binding to LMP1-CTAR2 and the new Q-E exchange experiments at P₀ of CD40 and RANK (for the latter see the new Figure 1h and Results). As suggested by this reviewer, we have cloned and tested the Y₃₈₄D mutation for its interaction with TRAF6 (new Supplementary Figure 1f and Results lines 200-209). The Y₃₈₄D mutation weakened the TRAF6 interaction with LMP1, but did not abolish it. This was anticipated and further confirms the new TRAF6 interaction consensus PxQ/ExxF/Y/D/E. This finding also suggests that the presence of D₂₀₉ within the putative TRAF6 motif of CTAR1 is most likely not the cause for the missing TRAF6 interaction. We therefore suggest that “additional factors adjacent to the putative TRAF6 core motif of CTAR1 might thus prevent TRAF6 interaction” (lines 208-209). We would like to add here that the focus of this manuscript has been the characterization of TRAF6 interaction with CTAR2. More detailed studies on CTAR1, which are clearly valid and interesting, are beyond the scope of the present manuscript, especially because TRAF6 does not interact with CTAR1 directly.

2. In figure 3, TRAF6 wildtype, but none of the binding-defective mutants, was able to rescue CTAR2 signaling to NF- κ B. How about LMP1 A204xAxA/Q381E and A204xAxA/T384F mutants, which showed enhanced affinity to TRAF6?

Response: We thank this reviewer for this suggestion, which brings up the question whether the interaction of the binding-defective TRAF6 mutants with LMP1 might have a threshold determined by the affinity of LMP1 to TRAF6. To be able to more precisely and directly quantify the effect of the high-affinity LMP1 mutation Q₃₈₁E (chosen as representative for LMP1 high-affinity mutations) on interaction with the binding-defective TRAF6 mutants R₃₉₂A and F₄₇₁A (the latter chosen as representative for TRAF6 residues interacting with P₀ of TRAF6), we decided to perform AlphaScreen

interaction assays instead of NF-kappaB reporter assays. We clearly see that also “high-affinity” LMP1 Q₃₈₁E is unable to override the negative effects of TRAF6 R₃₉₂A or F₄₇₁A mutation on TRAF6 interaction with LMP1 (new Supplementary Figure 2a and Results lines 229-231). This also means that the residues R392, F471 (and Y473) are in fact absolutely critical contacts between LMP1 and TRAF6.

In the revised version of the manuscript we also include RANK as an additional receptor for our interaction studies. These new experiments demonstrate that the molecular architecture of the LMP1-TRAF6 complex shows close similarities to RANK-TRAF6 (see below, the new Figures 1f, 1h, and 2c, and Results). RANK shows an affinity to TRAF6 of 7.7 nM, which is even higher than that of LMP1 Q₃₈₁E (14.7 nM). Supporting our conclusions for LMP1, TRAF6 R₃₉₂A and F₄₇₁A do not interact with RANK either (new Figure 2c).

3. The author claimed that “the TRAF6 mutant R392A revealed a striking difference regarding LMP1 and CD40 binding”. However, it is not so striking if LMP1 is compared with RANK. It is already shown that RANK interaction was not impaired by the TRAF6 mutant R392A (Nature volume 418, pages443–447 (2002)). The finding is rather exaggerated here.

Response: Despite the repeated in-depth study of the paper by Ye et al. (*Nature* **418**, 443-447 (2002)), we did not find any experimental data in this publication that address the effect of R₃₉₂A mutation on TRAF6 interaction with RANK. Interpreting their RANK-TRAF6 and CD40-TRAF6 crystal structures, the authors discuss possible interactions of R392 with RANK and CD40 as follows: “An amino-aromatic interaction is observed between Tyr 349 of TRANCE-R and Arg 392 of TRAF6. Structurally, a similar interaction should be possible for either Phe 238 of CD40 or an acidic residue, which is present in mouse CD40 (Fig. 1g)”. These observations rather imply a positive role for R392 in TRAF6 interaction with both RANK and CD40. However, no experiments were included in this publication, which directly address the relevance of R392 for TRAF6 interaction with RANK and CD40. Only one experiment involving R₃₉₂A has been presented by Ye et al., in which they tested the effect of this mutation on the potency of dominant-negative TRAF6 in IL-1 (IRAK) signaling (Fig. 2d in Ye et al.). Here, they saw a diminished dominant-negative activity of TRAF6.DN if R392 was mutated, which pointed to an important role for R392 in TRAF6 interaction with IRAK. Taken together, the paper by Ye et al. rather suggests important functions for R392 in TRAF6 binding to CD40, RANK, and IRAK. It was not shown in this paper that “RANK interaction was not impaired by the TRAF6 mutant R₃₉₂A”. We are therefore convinced that our finding that R392A discriminates between LMP1 and CD40 is novel and interesting, because it reveals structural differences between LMP1-TRAF6 and CD40-TRAF6 at the molecular level.

Stimulated by the comments of this reviewer (see also response to comment #5), we have now included RANK as a third receptor into our interaction experiments (new Figures 1f, 1h, and 2c) to compare RANK with LMP1 and CD40. We found that LMP1 and RANK are in fact very similar with

regard to their interaction mode with TRAF6, and both receptors differ from CD40. We find that LMP1 and RANK require R392 of TRAF6 and both receptors tolerate a Q-E exchange at P₀. However, LMP1 remains unique in the sense that it is the only known TRAF6-interacting protein that naturally carries a Q at P₀. In light of our new data we have rewritten large parts of the Results describing Figures 1, 2 and 4, which also led to the deletion of the sentence “*the TRAF6 mutant R392A revealed a striking difference regarding LMP1 and CD40 binding*”. For more details see also our answer to comment #5 of reviewer #1.

4. The molecular model of the LMP1-TRAF6 complex is proposed without any explanation of the starting complex or quality of the resultant model. Why did the author choose the MAVS-TRAF6 complex that is introduced rather abruptly? Wouldn't it be better to start from the CD40-TRAF6 complex (1LB6), especially when the authors compare that structure of the LMP1-TRAF6 complex with the CD40-TRAF6 complex? In addition, how was energy minimization conducted, and how was the quality of the minimized structure assessed?

Response: We acknowledge these are valid points. As we discuss in the revised version of the manuscript, our new functional studies show that LMP1 does not behave like CD40 but it behaves similar to RANK (Figures 1f-1g and 2a-2c) and Results. The observations from our functional studies are consistent with the differences observed in the X-ray structures of CD40-TRAF6 and RANK-TRAF6 (Ye et al., *Nature* **418**, 443-447 (2002)). Hence, it was obvious that RANK-TRAF6 is the more suitable template for modelling than CD40-TRAF6. We repeated the LMP1-TRAF6 modelling starting from the RANK-TRAF6 structure PDB 1LB5 (Ye et al., *Nature* **418**, 443-447 (2002)), see new Figure 4. We explain in the text why we are (now) using RANK-TRAF6 as a starting point for the modelling (Results, lines 285-288).

We agree with this reviewer that the initial description of the modelling process was not entirely clear and we have revised the manuscript accordingly. The modelling process is now explained in more detail in Methods (lines 695-708). In addition, we have revisited the model building process and rebuilt the LMP1-TRAF6 model from scratch derived from RANK-TRAF6 1LB5 using the Schrödinger platform, an industry-leading software platform for drug discovery. We are pleased to find that the new model is consistent with the previous model derived from MAVS-TRAF6.

The new model is in fact straightforward, based upon local modifications of the X-ray structural data of the RANK-TRAF6 complex. In other words, the model sticks very closely to X-ray structural data of RANK-TRAF6 and was derived according to the following protocol:

- We downloaded the structural data of the template structure (1LB5) from the PDB.
- We preprocessed the structure with the default protein preparation procedure of Schrödinger (all settings default). This is to correct any structural issues such as missing atoms in amino acid side chains, and to assign protonation states and add hydrogen atoms.

- We used the Schrödinger "mutation" feature to replace the relevant side chains of MAVS (or RANK) to match the sequence of LMP1. The important fact to know here is that the side chain orientations of the particular amino acid side chains in question can be derived with high confidence. This is because the structural properties of this particular protein-peptide complex leaves very few (if any) options for side chains to adopt different conformations.
- We refined the orientation of the mutated amino acid side chains by energy minimization with the OPL4 force field.

Because of the fact that this model sticks that closely to the experimental data and their interpretation we have high confidence in its validity.

5. The author claims that “the unique conformation of the LMP1-TRAF6 interface at P1 to P3 enables additional stabilizing contacts between LMP1 and TRAF6, which are not present in CD40-TRAF6, and may well explain why LMP1 tolerates glutamine at P0.” The interaction is not unique to the LMP1-TRAF6 interface, and a similar interaction was observed in the RANK-TRAF6 interaction as well as in the MAVS-TRAF6 interaction. One easy experiment to support the claim for glutamine toleration is to see if the interaction between RANK and TRAF6 tolerates the E to Q mutation. Furthermore, the structural model cannot explain why the LMP1 Y384F mutation improved the affinity. From the author’s claim, the LMP1 Y384F mutation disrupts the P1-P3 interface and makes the protein would not allow to have Q in the P0 position and thus cannot bind. These considerations indicate that the structural model is rather questionable and is prejudiced.

Response: As suggested by this reviewer we have now included RANK into our AlphaScreen interaction studies with TRAF6. To be able to present a coherent AlphaScreen PPI dataset including RANK and the mutants suggested by this reviewer, we have replaced Figures 1F and 1G of the first version of this manuscript by the newly generated PPI experiments now presented in the new Figures 1f to 1i, Figure 2c, and Supplementary Figures 1d and 1f. In fact, using our sensitive and quantitative TRAF6 interaction assay, we now find that both RANK and CD40 tolerate the exchange of E at position P₀ by Q (new Figures 1f and 1h), in contrast to what has been published before for CD40 using peptide array experiments (Pullen, et al. *J Biol Chem* **274**, 14246-14254 (1999)). When we replace Q381 at P₀ of LMP1 by E, we generate a more active version of LMP1. Notably, the change in affinity to TRAF6 of all three receptors after Q-E or E-Q exchange, respectively, is in the same order of magnitude, which suggests that the structure of the TRAF6 complex around P₀ is similar between the three receptors. This is also reflected in our LMP1-TRAF6 model, as compared to the published RANK-TRAF6 and CD40-TRAF6 structures (new Figure 4b). This finding makes a potential relevance of the LMP1-TRAF6 interface at P₊₁ to P₊₃ for Q toleration at P₀ obsolete, because all three receptors can tolerate Q, which is a new finding. The potential reason why LMP1 carries a Q at P₀, in contrast to all known cellular TRAF6-interacting proteins, is discussed in the Discussion part (lines 437-443). We have re-written the interpretation of our model in the light of these new findings and are confident that

our interpretation is now sound and in line with published data and structures (Text Results to Figure 4).

We cannot fully explain the increased affinity of LMP1 Y384F to TRAF6 from the model, which is, to our opinion, no fundamental problem of our LMP1-TRAF6 model, especially because the effects of all other mutations can be perfectly explained by the model. We have revisited the Y384F effect and now suggest that “The observed increase in affinity of LMP1 Y₃₈₄F towards TRAF6 might be related to the possible formation of hydrophobic interactions with TRAF6, which are not possible with a tyrosine at P₃” (see lines 332-334).

6. The author claims that “Upon addition of the LMP1 peptide, the peaks corresponding to these residues are broadened beyond detection, indicating that these residues contribute strongly to binding.” This is a misleading statement. The signal broadening would not indicate the contribution to the binding. It indicates that these residues are within the region affected by the interaction. In addition, the signal broadening indicates the presence of multiple conformations in the complex, which poses a question to the correctness of discussing with a single static conformation from the binding model.

Response: The disappearance of peaks on an HSQC spectrum indicates the so-called intermediate exchange regime. This is the case when complex lifetime is comparable to the length of single NMR scan (typically few hundred msec). In this case, an NMR peak is composed partially of two split peaks (slow exchange regime) and single peak shifting (fast exchange regime, <https://nmr.chem.ucsb.edu/education/part3.html>). Practically, such averaging leads to severe broadening of the resonance and its disappearance in the noise. It is not correct to attribute signal disappearance to multiple conformations (as is the case in electron density of the X-ray). Multiple, stable conformations are seen on the HSQC spectrum as separate sets of weaker peaks. While it is true that this is not a direct evidence of binding, the chemical shift perturbation is caused by altering chemical environment of the observed nuclei. Since the only difference between samples is a presence of LMP1 peptide, it is safe to attribute the change of chemical environment (structure) to the LMP1 binding. Allosteric change in the protein would cause much more extended chemical shift changes on the entire spectrum.

We agree with the reviewer that the expression “ *these residues contribute strongly to binding* ” might be misleading in terms of attributing energy contribution to the residues with strongest chemical shift (which is often the case but it is not the rule).

Therefore, we altered the text (lines 346-356) to:

“Upon addition of the LMP1 peptide, the peaks’ resonance corresponding to some of the TRAF6 residues were broadened beyond detection, which is indicated by the appearance of the blue resonance of free TRAF6 (Figure 5a). This is an indication of a significantly altered chemical environment in the

so-called intermediate exchange regime when the protein-ligand complex lifetime is comparable to the length of a single NMR scan. This behavior is often observed by "anchor" residues of the protein-protein interaction where the residues responsible for most binding energy tend to have much slower bound/unbound exchanges than the residues contributing less. Their peaks shift after the addition of the TRAF6 peptide. Additionally, a clear pattern of other resonances shifting upon the addition of the LMP1 peptide was observed, proving direct binding. The chemical shifts observed are localized in the close proximity of the predicted LMP1 binding pose and are in line with our biochemical and molecular modeling data (Figure 5b)."

7. The CSP mapping poorly matches the proposed model as it extends horizontally in the figure. Especially, the chemical shift mapping does not show any perturbation near the P3 site in the proposed model, while the residue strongly contributes to the interaction. The author might want to indicate the residues not assigned in the figure.

Response: We thank this reviewer for this suggestion. In the first version of our manuscript we have already indicated unassigned residues in Figure 5B, but apparently the color code was unclear. In the revised version of our manuscript we changed the color code of Figure 5b to indicate more clearly: assigned residues showing strong shifts (dark blue), weaker shifts (light blue), or no shifts (wheat), and TRAF6 residues, which have not been assigned at all (light grey). The CSP mapping is not a very precise method and, given that a significant part of the proposed binding site especially around P₊₃ is not assigned, we regard the correlation between our prediction and the CSP mapping as very good. Typically, a few residues away from the binding site will have large CSP, this is frequently seen. It is evident that, when one disregards unassigned residues, the CSP group around the predicted binding region.

Reviewer #2 (Remarks to the Author):

Gehler et al. describe much work to establish that TRAF6 binds the CTAR2 directly and thereby resolve a subject of uncertainty in the mechanism by which the LMP1 oncogene signals via TRAF6 in cells. They wish also to test the role of this binding in cell survival. They show that targeting both LMP1 and TRAF6 in two murine cell lines with CRISPR and gRNAs decreases their survival and acknowledge that this finding does not establish a role for the direct binding of TRAF 6 and LMP1 in the survival of these cells. They use cell permeable peptides to block this binding, somewhat unconvincingly to test their contention directly. For example, the data for the cell line HA-LCL3 in Fig. 7E is either misleading or indicates that 90% of the cells are killed by the control peptide.

Response: We thank this reviewer for this valuable hint. The presentation of the data for HA-LCL3 in the previous Figure 7E was indeed misleading, because the cell viability data (MTT conversion) were given on the y-axis as OD values (direct output from the reader), which differ in their levels from cell line to cell line. Therefore, the viability data were not directly comparable between the cell lines, which was not explicitly stated. We repeated the complete experiment to include LCL.NGFR-LMP1 cells as a third LCL, and now provide cell viability data as relative percentages. Cell viability in the presence of non-targeting control peptide was set to 100 % for each cell line and viability in the presence of TRAF6 inhibitor peptide was expressed for each cell line as the percentage relative to the control peptide. The new data are included in the new Figure 7h (previously 7E) and demonstrate (as already our previous data) that the TRAF6 inhibitor peptide strongly interferes with LCL, but not Burkitt, survival.

The authors need to provide some other approach to demonstrate a need for direct binding for the survival of these cells. For example, can they express CTAR2-null LMP1 with a coiled-coil domain and TRAF6 with its partner coiled-coil domain so that they can only interact with each other (see: Protein Sci. 2012 Apr; 21(4): 511–519) to support LMP1's functions? Some positive test for a role for direct binding in cell survival is needed to establish the authors contention.

Response: We thank this reviewer for the interesting suggestion of an experiment, which would provide direct positive prove of the requirement of the direct TRAF6 interaction with LMP1 for LCL survival. However, to our knowledge, the suggested coiled-coil tag system published by Fernandez-Rodriguez et al. in 2012 has only been used for in vitro studies so far, but not for the enforced interaction of proteins within cells. It is, thus, unclear whether this system is really working within cells and suitable for the suggested in vivo experiment in LCLs. Moreover, it would have been technically very demanding to establish LCLs that only express coiled-coil tagged LMP1deltaCTAR2 and coiled-coil tagged TRAF6, but not the wild-type proteins.

Instead and to address the valid criticism of this reviewer, we achieved enforced TRAF6 recruitment to CTAR2-null LMP1 by generating a fusion construct of LMP1, lacking CTAR2, with wild-type TRAF6, both separated by a flexible linker (see the new Figures 7a to 7c). This construct mimics constitutive interaction of TRAF6 with LMP1-CTAR2 and is able to activate CTAR2-like signaling in TRAF6-KO

MEFs (in the absence of functional CTAR1, new Figure 7b). LCL.NGFR-LMP1 cells were transduced with an inducible lentiviral vector expressing the LMP1(delta371-386)-liTRAF6 fusion. In the LCL experiment, the fusion construct was used with a functional wild-type CTAR1 because it is well established in the literature that LCL survival depends on both, CTAR1 and CTAR2. To switch off endogenous wild-type LMP1 activity, the cells were deprived from cross-linking antibodies and expression of the fusion protein was induced. The LMP1(delta371-386)-liTRAF6 fusion construct and, thus, the enforced direct TRAF6 recruitment to LMP1deltaCTAR2 was sufficient to support survival of the LCLs (new Figure 7c). This new positive test establishes the important role of direct TRAF6 binding to LMP1 for LCL survival.

Reviewer #3 (Remarks to the Author):

Giehler et al. provided new insight of LMP1-TRAF6 signaling. The authors showed that direct interaction of TRAF6 to LMP1. They analyzed the interaction site of TRAF6 on the intercellular part of LMP1. Finally, they showed that the direct interaction of TRAF6 to LMP1 is critical for NF- κ B activation and survival of LMP1-driven B cell lymphoma.

Although the current study contains interesting points in the field of TRAF biology, the direct interaction of TRAF6 and LMP1 is not novel finding. The function of LMP1-TRAF6 was already studied at 2012 with several studies and the direct interaction of LMP1 to TRAF6 was also analyzed at 2011.

- 1. Molecular mechanisms of TNFR-associated factor 6 (TRAF6) utilization by the oncogenic viral mimic of CD40, latent membrane protein 1 (LMP1) J Biol Chem 2011 Mar 25;286(12):9948-55*
- 2. TRAF6 is a critical regulator of LMP1 functions in vivo Int Immunol. 2014 Mar;26(3):149-58.*
- 3. TRAF binding is required for a distinct subset of in vivo B cell functions of the oncoprotein LMP1 J Immunol. 2012 Dec 1;189(11):5165-70. doi: 10.4049/jimmunol.1201821.*

Response: We thank this reviewer for the statement that our manuscript provides new insight into LMP1-TRAF6 signaling and the role of this direct interaction in LMP1-driven signaling and lymphoma development. We would like to add here that we also characterize this interaction as a novel potential target for intervention by inhibitory molecules.

We fully agree that, and this fact has been widely discussed already in the first version of our manuscript, TRAF6 functions in LMP1 signaling have been studied previously. Publication #1 mentioned by this reviewer (Arcipowski et al., J Biol Chem **286**, 9948-9955 (2011), Ref. 22) has been cited and discussed in lines 74-75 (presumably indirect interaction of TRAF6 with CTAR1), lines 77-78 (role of TRAF6 in LMP1 CTAR2 signaling), lines 96-98 (TRAF6 lacking its TRAF domain is unable to rescue LMP1 signaling), lines 451-452 (TRAF6 and LMP1 observed in one signaling complex), line 473 (TRAF6 has a role in CTAR1 signaling) and lines 474-476 (TRAF6 co-immunoprecipitates with CD40-LMP1 from mouse B cells). Publication #2 (Arcipowski et al., Int Immunol **26**, 149-158 (2014), Ref. 52) has been discussed in lines 102-103 (TRAF6 deficiency affects B cell numbers driven by CD40-LMP1 in lymph nodes of transgenic mice). Publication #3 (Arcipowski et al., J Immunol **189**, 5165-5170 (2012)) is now included as Ref. 51. This publication was not cited initially because it describes the role of the CTAR1-TRAF binding site in B cell functions, but does not directly relate to TRAF6.

Here, we would like to point out that also earlier reports by our and other groups described functions for TRAF6 in CTAR1 and CTAR2 signaling, which have been cited in the manuscript and discussed in the context of our new observation that CTAR2 and TRAF6 interact directly. In fact, it was our laboratory who originally discovered the important role of TRAF6 in LMP1 signaling (Schultheiss et al., EMBO J **20**, 5678-5691 (2001), Ref. 20). In this publication we also found TRAF6 in NGFR-LMP1

signaling complexes in vivo for the first time and showed that TRAF6 interaction is mediated by CTAR1 and/or CTAR2.

However, and most importantly, none of the previous studies, including our own previous studies and the three publications listed by this reviewer, were sufficient to clarify the molecular mode of TRAF6 interaction with LMP1. Co-immunoprecipitations (as for instance performed by Arcipowski et al., J Biol Chem **286**, 9948-9955 (2011), Ref. 22), pull-downs from cell lysates or in situ immunofluorescence microscopy studies are suitable methods to demonstrate the presence of two proteins in one complex but they are insufficient to prove a direct protein-protein interaction, because the interaction could still be mediated by other factors present within the cell or lysate.

To demonstrate a direct PPI, two-component systems of the two proteins of interest are required, which exclude a potential role of third factors for the interaction. To this end, we have established several two-component systems in the present manuscript to demonstrate and study the direct interaction between LMP1 and TRAF6, see below and Figures 1a to 1i, Supplementary Figure 1, Figures 2a to 2c, Supplementary Figure 2a, Figure 5, and Figures 7e to 7g. Our data generated with these systems clearly show that TRAF6 interacts directly with CTAR2, but not with CTAR1 (Figures 1b, 1c, 1e, 1f, and 1i). We discuss the possibility of an indirect recruitment of TRAF6 to CTAR1 and suggest a possible mechanism in the Discussion section (lines 473-484).

The evidence of direct interaction between LMP1 (suggested peptide) and TRAF6 is still vague. Suggest to perform SPR and ITC (peptide/protein interaction).

Response: We acknowledge the suggestion of detecting the direct interaction of LMP1 peptide with TRAF6 by SPR or ITC. However, we are convinced that we have already firmly established the direct interaction between LMP1 and TRAF6 in the manuscript by the following independent two-component systems:

1. In vitro binding assays of purified recombinant GST-LMP1(181-386) (complete LMP1 signaling domain fused to GST) and purified recombinant His-TRAF6(310-522) (complete CC and TRAF-C domains). The direct TRAF6 binding to LMP1 was dependent on Y384 but not on CTAR1. Of note, these experiments were performed with full proteins, not just peptides. See Figures 1a and 1b, Supplementary Figure 1.
2. Binding of purified recombinant His-TRAF6(310-522) (or TRAF2) to LMP1- and CD40-derived peptides, immobilized on filter arrays. See Figure 1c and Supplementary Figure 1c.
3. We established an AlphaScreen-based LMP1-TRAF6 protein-protein-interaction assay, which is also suitable for high throughput screening, composed of purified recombinant GST-LMP1(181-386) (complete LMP1 signaling domain) and purified recombinant His-TRAF6(310-522) (CC and TRAF-C domains). This assay is able to detect and quantify the interaction between the two full proteins. With this assay, we have also measured K_D values for the direct interaction between LMP1 (and RANK,

CD40) with TRAF6. For the assay principle see Figure 1d, for the assay results see Figures 1e to 1i, 2a to 2c, 7e to 7g, and Supplementary Figures 1d, 1f and 2a.

4. NMR experiments, in which the binding of an LMP1 peptide to purified recombinant TRAF6 was detected by the clear shifting of the resonances of some TRAF6 residues upon peptide interaction, a clear sign of direct interaction. NMR is an accepted method to prove the interaction of a peptide with a protein and is comparable in its significance to SPR and ITC. See Figure 5.

Because these four methods, to our opinion, already provided ample and sufficient experimental prove to claim a direct interaction between LMP1 and TRAF6 and because the AlphaScreen assays also delivered quantitative interaction data, we decided to abandon the laborious establishment of SPR or ITC as a fifth (or sixth) method, which would not deliver significant additional information.

The complex structure modeling was not perfect. the validation should be done. Structure modeling (peptide/protein) contains limited information.

Response: Stimulated by the comments of this reviewer and reviewer #1, as well as by our new results of PPI experiments revealing similarities between LMP1 and RANK with respect to their interaction with TRAF6, we repeated the LMP1-TRAF6 modeling starting from the RANK-TRAF6 crystal structure PDB 1LB5 with essentially identical results as our previous modeling based upon the MAVS-TRAF6 crystal structure. This makes us confident that the model is correct (see new Figure 4 and response to reviewer #1 for more details). Our new experimental and modeling data suggest that LMP1-TRAF6 and the RANK-TRAF6 have comparable structures, which differ from the CD40-TRAF6 structure. We have added a new set of experimental AlphaScreen interaction data of LMP1, RANK and CD40 including new mutants (Figures 1f to 1i, 2a to 2c), which are in line with and, thus, further validate our LMP1-TRAF6 model. The interpretation and discussion of the model has been revised accordingly (see Results).

REVIEWERS' COMMENTS

Reviewer #1 (Remarks to the Author):

The reviewer confirms that the authors adequately conducted additional mutational experiments, interaction experiments, and modeling according to comments #1-5 & 7 and satisfactorily revised the manuscript. However, there is a critical mistake in interpreting NMR data in response to comment #6. As the experimental NMR spectroscopist, this cannot be overlooked.

In the response, the authors basically claimed that the disappearance of peaks on an HSQC spectrum indicates the so-called intermediate exchange regime between "free and bound" states. However, this is not the case here. In the experimental condition, a 5-fold amount of LMP1 peptide was added to 90 μ M TRAF6, and the binding between them was 72 nM. In such a case, the binding is fully saturated, and no exchange between the free and bound states is present in the sample. Thus, regardless of the exchange rate between the free and bound states, we will see the appearance of the bound signals upon the titration of the peptide.

Thus, if "the peaks' resonance corresponding to some of the TRAF6 residues were broadened beyond detection" in the bound condition, it should be interpreted as there are multiple (more than 2) bound conformations that are exchanging in the intermediate time scale.

Thus, the authors did not fully answer the previous comment #6 and the reviewer's concern about the correctness of the discussion based on a single static conformation from the binding model.

Reviewer #2 (Remarks to the Author):

Giehler et al. have addressed my concerns well.

Reviewer #3 (Remarks to the Author):

In this paper, the reliability of the data is very important, and it is believed that the reliability of the data has been increased through several additional experiments. However, there is still a disadvantage of not using the most common method used for most TRAF-receptor peptide binding analysis

Response to the reviewers:

Reviewer #1 (Remarks to the Author):

The reviewer confirms that the authors adequately conducted additional mutational experiments, interaction experiments, and modeling according to comments #1-5 & 7 and satisfactorily revised the manuscript.

Response: Thank you very much!

However, there is a critical mistake in interpreting NMR data in response to comment #6. As the experimental NMR spectroscopist, this cannot be overlooked.

In the response, the authors basically claimed that the disappearance of peaks on an HSQC spectrum indicates the so-called intermediate exchange regime between “free and bound” states. However, this is not the case here. In the experimental condition, a 5-fold amount of LMP1 peptide was added to 90 μ M TRAF6, and the binding between them was 72 nM. In such a case, the binding is fully saturated, and no exchange between the free and bound states is present in the sample. Thus, regardless of the exchange rate between the free and bound states, we will see the appearance of the bound signals upon the titration of the peptide.

Thus, if “the peaks’ resonance corresponding to some of the TRAF6 residues were broadened beyond detection” in the bound condition, it should be interpreted as there are multiple (more than 2) bound conformations that are exchanging in the intermediate time scale.

Thus, the authors did not fully answer the previous comment #6 and the reviewer’s concern about the correctness of the discussion based on a single static conformation from the binding model.

Response:

We are happy to provide further clarification. We would like to point out that the observation that we could not detect the bound state even at 5-fold molar excess, for a K_D in the nanomolar range may reflect unusual binding kinetics, for example, with a slow not diffusion controlled on-rate, which could indicate more complex binding. We agree that we cannot exclude the possibility of additional contributions to line-broadening from dynamic processes related to potential multiple (slightly different) local contacts (with different chemical shifts of the bound state), or conformational dynamics of the binding site. We have clarified this in the text accordingly.

Reviewer #2 (Remarks to the Author):

Giehler et al. have addressed my concerns well.

Response: Thank you very much!

Reviewer #3 (Remarks to the Author):

In this paper, the reliability of the data is very important, and it is believed that the reliability of the data has been increased through several additional experiments. However, there is still a disadvantage of not using the most common method used for most TRAF-receptor peptide binding analysis.

Response: Thank you very much for the acknowledgment that the reliability of the data has been increased in the revised version of the manuscript.

As already stated in our response to the comments of this reviewer to the first version of the manuscript, we have applied multiple independent and state-of-the-art two-component assays to demonstrate the direct interaction of LMP1 (peptide and full protein) with TRAF6, including AlphaScreen PPI assays with purified full proteins, NMR, and others. To our opinion, all these assays in combination are fully sufficient to demonstrate the direct interaction of both proteins. Nevertheless, we acknowledge that SPR or ITC could be useful alternative methods for this purpose.